

# CMIP7 Data Request: Ocean and Sea Ice Priorities and Opportunities

Baylor Fox-Kemper[1], Patricia DeRepentigny[2], Anne Marie Treguier[3], Christian Stepanek[4], Eleanor O'Rourke[5], Chloe Mackallah[6], Alberto Meucci[6,7], Yevgeny Aksenov[8], Paul J. Durack[9], Nicole Feldl[10], Vanessa Hernaman[11], Céline Heuzé[12], Doroteaciro Iovino[13], Gaurav Madan[14,15], André L. Marquez[16], François Massonnet[2], Jenny Mecking[8], Dhrubajyoti Samanta[17], Patrick C. Taylor[18], Wan-Ling Tseng[19], Martin Vancoppenolle[20]

[1]Department of Earth, Environmental, and Planetary Sciences (DEEPS), Brown University, Providence, Rhode Island, 02912, USA
[2]Earth and Climate Research Center (ELIC), Earth and Life Institute (ELI), Université catholique de Louvain (UCLouvain), Louvain-la-Neuve, Belgium
[3]Laboratoire d'Océanographie Physique et Spatiale, University of Brest, CNRS, Ifremer, IRD, Brest, France
[4]Alfred Wegener Institute - Helmholtz Center for Polar and Marine Research, Bremerhaven, Germany
[5]CMIP International Project Office, ECSAT, Harwell Science & Innovation Campus, UK
[6]Climate Science Centre, CSIRO Environment, Aspendale, VIC, Australia
[7]Department of Infrastructure Engineering, The University of Melbourne, Melbourne, VIC, Australia
[8]National Oceanography Centre, Southampton, UK
[9]PCMDI, Lawrence Livermore National Laboratory (LLNL), Livermore, California, 94550, USA
[10]Department of Earth and Planetary Sciences, University of California, Santa Cruz, Santa Cruz, California, USA
[11]Commonwealth Scientific and Industrial Research Organisation, Aspendale, Victoria 3195, Australia
[12]Department of Earth Sciences, University of Gothenburg, Gothenburg, Sweden
[13] Foundation Euro-Mediterranean Centre on Climate Change (CMCC), Bologna, Italy
[14]National Centre for Atmospheric Science, University of Reading, Reading, United Kingdom
[15]Section for Meteorology and Ocean Sciences, University of Oslo, Oslo, Norway
[16]Department of Earth System Numerical Modelling (DIMNT), National Institute for Space Research (INPE), Sao José dos Campos, Sao Paulo, Brazil
[17]Earth Observatory of Singapore, Nanyang Technological University, Singapore
[18]National Aeronautics and Space Administration (NASA), Langley Research Center, Hampton, Virginia, 23681, USA
[19]Ocean Center, National Taiwan University, Taipei, Taiwan
[20]Laboratoire d'Océanographie et du Climat, CNRS/IRD/MNHN, Sorbonne Université, Paris, France

*Correspondence to*: Baylor Fox-Kemper (baylor@brown.edu)

**Abstract.**

The ocean and sea ice are central to Earth's climate system, influencing global heat and carbon cycles, weather patterns, and sea level rise. Recent decades have seen rapid advances in Earth System Models (ESMs), but limitations remain in simulating and comparing key oceanic and cryospheric processes across models. A recurring challenge in model intercomparison efforts like the Coupled Model Intercomparison Project (CMIP) is determining the output variables that best represent essential mechanisms while remaining manageable in volume and complexity. Here we present the CMIP7 ocean and sea ice data request, developed through an international, community-based process to prioritize



variables for model output. We identify seven *opportunities*—science-based use cases spanning ocean and cryosphere
    drivers and responses, paleoclimate, polar amplification, extremes, wind waves, and rapid model evaluation—to guide
    variable selection and temporal resolution. To address these opportunities we request new high-frequency and depth-
    integrated variables, support improved diagnostics of ocean heat uptake, sea ice processes, and model-observation
    comparison, and build on lessons from CMIP6. Our approach enables targeted, efficient, and transparent data curation
to support a wide range of users, from model developers to policymakers. This effort reflects a growing need for more
    sophisticated, integrative model outputs that address pressing climate questions, including regional extremes and
    tipping points, while laying the groundwork for future modeling developments.

## 1 Introduction

The ocean and sea ice play several critical roles in the Earth system (Fox-Kemper et al., 2021a). One of the most well-
known is the oceans' capacity to act as a vast reservoir for thermal energy: since the 1950s, over 90% of the excess
    energy on Earth resulting from human activities has been stored in the oceans (Johnson & Lyman, 2020; Cheng et al.,
    2022; Johnson et al., 2022; Li et al., 2023). Similarly, the ocean takes up about a quarter of the anthropogenic carbon
    emissions, resulting in ocean acidification (e.g., Gruber et al., 2023). Oceans cover about 72% of Earth's surface, are
    the source of most of the evaporated water, and receive most of the precipitation that falls back to the surface (e.g.,
Mayer et al., 2021). The ocean contributes about a third of the meridional heat transport from the equator to the poles,
    with the remainder divided fairly evenly between the latent heat transport of the water cycle (poleward via humidity
    and equatorward as liquid ocean water) and the atmosphere (Trenberth, 2022). Furthermore, the ocean participates in
    many coupled modes of variability with global relevance, such as the El Niño-Southern Oscillation (ENSO). However,
    due to their vast mass and thermal capacity, the ocean adjusts more slowly to changes than the atmosphere does,
causing it to lag behind the atmosphere in response to external forcing (Frankignoul & Hasselmann, 1977). This
    capacity enables the oceans to buffer transient climate changes to some extent, dampening and delaying the full effect
    of external forcing on the climate system (e.g., Stuecker, 2023). Finally, changes in ocean conditions directly affect
    human society through local climate impacts (e.g., land-sea breezes, monsoons, and marine climate), sea level rise,
    coastal inundation and erosion, and shifts in marine resources such as fisheries and transportation (Cooley et al., 2022).
Although sea ice constitutes only a small fraction (about 0.1%) of Earth's total ice volume, it has many consequential
    climate effects (Fox-Kemper et al., 2021a). The combined Arctic and Antarctic sea ice systems cover an area ranging
    between 16 and 28 million km², depending on the time of the year, corresponding to about 4-8% of the global ocean
    surface. Sea ice affects the albedo of the Earth, insulates the oceans from the atmosphere, and is an important habitat
    for many species. The formation and melting of sea ice affects the formation of key ocean water masses (e.g.,
Abernathey et al., 2016). There is a long-running debate about whether sea ice affects mid-latitude extreme weather



(e.g., Francis, 2017; Screen et al., 2018). The polar oceans and sea ice also impact land ice by interacting with the ice shelves that buttress the ice sheets (Sun et al. 2020; Reese et al., 2023; Bradley and Hewitt, 2024). Finally, the reduction of sea ice, both in terms of areal coverage and volume, is one of the clearest indicators of ongoing climate change. In February 2025, global sea ice coverage reached a record low, with Arctic sea ice 8% below average and

Antarctic sea ice 26% below average (v2.2 data based on Lavergne and Down, 2023).

These considerations highlight the value of a careful stocktake and selection of ocean and sea ice characteristics critical for our understanding of a changing climate has for supporting research over the coming years. Our team assembled experts from 21 institutions to discuss and prioritize the variables in modern Earth System Models (ESMs) most relevant for study of the oceans and sea ice. ESMs, including those participating in the upcoming Coupled Model

Intercomparison Project Phase 7 (CMIP7), are designed to simulate many of the effects described above. The multiple climactic roles of ocean and sea ice require a variety of variables to accurately quantify their interactions and their tendencies, across multiple frequencies, timescales, depth ranges, background climates and forcing scenarios. This paper aims to identify and prioritize the key ocean and sea ice data variables to be requested from ESMs, facilitating comparison among models and with observations, revealing mechanisms, and monitoring changes. Selecting variables

for data requests requires careful judgment: 1) excessive data demands can overwhelm the capabilities of modeling centers, users, and storage facilities and increase the risk of inefficiencies and errors in data management, 2) structures must be imposed for the timely release of data and ensuring seamless workflow integration adhering to deadlines from higher level activities, including Intergovernmental Panel on Climate Change assessment reports (e.g., IPCC, 2021, and the upcoming Seventh Assessment Report), and 3) a broad, inclusive user community is desired. The data request

decisions must be reasoned and judicious.

The data generated in CMIP7 Assessment Fast Track (CMIP7 AFT) and the rest of CMIP7 and related model intercomparisons will serve multiple user groups, including modelers aiming to improve their products (e.g., Fox-Kemper et al., 2019), observationalists seeking context for past and present measurements, as well as scientists, policy-makers, and managers evaluating the future impacts of ocean and climate changes on vulnerable natural and built

systems. To address their diverse needs, the Ocean and Sea Ice Author Team, under the wider CMIP7 Data Request Task Team (Mackallah et al., 2025), has identified a number of *opportunities* that represent both traditional and new applications for model data and that motivate the choice of *physical parameters*, *variables*, and *variable groups* requested. These opportunities are selected based on their potential to enhance understanding of the roles of the oceans and sea ice in the climate system and their projected changes.

This paper introduces the opportunities related to ocean and sea ice, lays out the related groups of variables, with special attention to variables to be requested for the first time for CMIP7 and their geographic and temporal sampling requirements, and clarifies variable definitions or provides references where they are carefully defined. It is not the role of this paper to elaborate on definitional choices, analyze the sensitivity of results to subtle differences in variable



definition, or prescribe the protocols needed for different model intercomparison exercises. Companion papers, such as
the Ocean Model Intercomparison Project (OMIP) for CMIP7 (Fox-Kemper et al., 2025, in preparation) and others
(Notz et al. 2016, McDougall et al. 2021, Treguier et al. 2023) serve to complement this paper and fulfill these
necessary roles. The accompanying tables of variables are archived as a dataset (CMIP Model Benchmarking Task
Team, 2024).

## 2 Approach and methodology

The Oceans and Sea Ice Author Team was recruited via open call between 2 February and 1 March 2024 (https://wcrp-
cmip.org/cmip7-ocean-seaice-call/). Members were sought from across the ocean and sea ice communities to gather
variable requirements for the CMIP7 Data Request, which is collaboratively organized using the platform Airtable.
Applications were reviewed by OMIP and Ocean Model Development Panel (OMDP) representatives alongside three
members of the Data Request Task Team. A diverse final group of 21 authors was formed, including World Climate
Research Program Core Project representatives from Climate and Cryosphere (CliC), Climate and Ocean Variability,
Predictability and Change (CLIVAR, including the Ocean Model Development Panel and its OMIP Working Group),
and Earth System Modeling and Observation (ESMO).  The team also included representatives from the following
Model Intercomparison Projects: Ocean Model Intercomparison Project (OMIP), Sea Ice Model Intercomparison
Project (SIMIP), Paleoclimate Model Intercomparison Project (PMIP), and the High Resolution Model Intercomparison
Project (HighResMIP). The author team spans a range of geographical regions, genders, career stages, and CMIP
experiences.

Many members of this team were not involved in previous data requests, but all were creators and/or users of past
CMIP or other ocean and sea ice model data. Consequently, new ideas and opinions are combined with the legacy of
CMIP6 and earlier CMIP rounds of data requests. Decisions on the variable selection involved reflecting on user
experiences in what was effective and what was not, and building on information about which variables were
downloaded and which were used for major publications and assessment reports such as the United Nations
Intergovernmental Panel on Climate Change Sixth Assessment Report (AR6: IPCC, 2021).

The team first convened on 28 June 2024, with community engagement activities beginning subsequently alongside the
first public consultation. Author team members were instructed to utilize their networks as community representatives
to gather scientific requirements for the ocean and sea ice components of the CMIP7 Data Request. Through the first
consultation phase, 11 opportunities were submitted with the initial selection of variables and their technical definitions
(Annex 1). The author team met every two to three weeks to discuss the submitted opportunities, identify any
remaining gaps, integrate input from the wider community, and focus on variable group development and refinement. A
harmonization sprint, involving all thematic teams, was held in September 2024, which resulted in the merging of



several opportunities within and across themes (the Ocean and Sea Ice Theme is one example), and culminated in the designation of themes to lead each opportunity. The Ocean and Sea Ice Theme progressed with seven opportunities after reviewing the output of the cross-thematic sprint and agreeing on appropriate merges of opportunities. The final list of opportunities led by the Ocean and Sea Ice Theme is found in Table 1. Further details of how the author team approached and conducted the decision making for each consultation phase can be found in Annex 1.

Following the v1.0 release in November 2024, the team focused on finalizing variable groups, supporting the processing of new variables and contributing to cross-theme meetings. Regular team meetings continued with additional sub-group meetings which focused on opportunity- or variable-specific requirements. GitHub discussions refined those opportunities requiring new CF standard names (https://github.com/cf-convention/vocabularies/issues) or other technical decisions. Collaborative spreadsheets helped to gather input between meetings, with some members of

the team interacting directly with the Airtable from a very early stage, with International Project Office (IPO) support and Data Request Task Team liaison members updating the Airtable records as needed. A systematic variable review was conducted during the Phase 2 and Phase 3 consultation periods to address comments, rectify errors and highlight remaining outstanding issues to the team. Furthermore, author team members contributed to cross-thematic meetings on issues such as consistent use of time subsets and separation, categorization, and prioritization of variable groups within

opportunities. Following the v1.1 release, opportunity proposers, who were not part of the existing author team, were invited to join to facilitate their contribution to paper development and final variable selection.

| ID | Opportunity Title | Variable Groups | Experiment Groups | Total number of variables |
|---|---|---|---|---|
| 47 | Ocean Changes, Drivers and Impacts | 19 | 4 | 240 |
| 73 | Sea Ice Changes, Drivers and Impacts | 14 | 3 | 191 |
| 51 | Paleoclimate Research at the Interface Between Past, Present, and Future | 13 | 4 | 266 |
| 13 | Causality of Polar Amplification | 6 | 2 | 88 |
| 49 | Ocean Extremes | 6 | 5 | 37 |
| 24 | Advancing Wind Wave Climate Modelling for Coastal Zone Dynamics, Impacts, and Risk Assessment | 5 | 3 | 132 |
| 68 | Effects and Feedbacks of Wind-Driven Ocean Surface Waves Coupled Within Earth System Models | 8 | 4 | 165 |
| 5 | Rapid Evaluation Framework | 1 | 2 | 20 (Oceans & |



| 5 | | | | Sea Ice only) |
|---|---|---|---|---|

**Table 1.** Data request opportunities led by the Ocean and Sea Ice Theme, including total number of variable groups, experiments requested, and variables.

## 3 Ocean & Sea Ice Opportunities included in the CMIP7 data request

The opportunities selected by the author team are presented roughly in order from the most familiar variables from previous data requests to those opportunities that require many new variables. Each opportunity description motivates some basic science questions, involves justification of frequency and resolution for which specific variables are needed, and presents some of the ideas behind newly introduced variables. Where relevant, these opportunities relate to some of the other CMIP7 themes, and these linkages are spelled out. Version 1.2.1 of the CMIP7 data request (Data Request Task Team, 2025b) provides all of the variables requested, not just the novel ones emphasized in this paper.

**Ocean Changes, Drivers and Impacts Opportunity (ID 47)**

As already noted, the ocean plays a vital role in the climate system by absorbing heat and carbon dioxide, regulating global temperatures, and influencing weather patterns. As the main source of uncertainty in seasonal to decadal (i.e., near-term) projections, the internal variability of the climate system is often stemming from ocean processes: ENSO, the Pacific Decadal Oscillation (PDO), the Atlantic Meridional Variability (AMV/AMO), etc. (IPCC, 2021, Annex IV). These modes of variability involve changes in ocean heat and salt content, sea ice properties, as well as transport by ocean currents at all depths. In this context, the goal of this opportunity is to continue efforts started in previous CMIP phases to quantify the processes that drive ocean variability and change and to provide understanding and a more robust assessment of climate projections (e.g., Griffies et al., 2016; Orr et al., 2017). In addition, this opportunity aims to better coordinate modeling efforts and comparison across models, for example, through improved grid specifications, as well as investigating the impacts of changes in oceanic properties on the global climate system.

The Meridional Overturning Circulation (MOC) is an important aspect of climate and an active part of the oceanic response to climate change. The Atlantic Meridional Overturning Circulation (AMOC) is projected to decline during this century (Fox-Kemper et al., 2021a), and a potential future collapse would imply a dramatic climate shift with enormous global and regional impacts (e.g., Zhang et al., 2019; Bellomo and Mehling, 2024). Many processes and feedback that govern AMOC are still debated, for example the role of warming versus freshwater forcing (Wen et al., 2023), input from Arctic sea ice and ice sheet melting (He and Clark, 2022), and the role of deep convection in different regions (Menary et al., 2020). Furthermore, the Southern Ocean MOC also plays a key role in sequestering heat and carbon (Williams et al., 2023). Quantifying and understanding the processes governing deep water formation and upwelling requires a full-depth analysis, considering the strong mesoscale variability (Morrison et al., 2016, Hewitt et al., 2020, Jackson et al., 2020). Within this opportunity, MOC-related variables are clustered together in the variable



group called *ocean_meridional_overturning_streamfunctions* to facilitate a better understanding of the changes in the MOC and their potential drivers.

Other processes and mechanisms prioritized in this opportunity include the relationship between changes in the ocean and polar processes and mechanisms, which remains poorly understood. For example: To what extent is ocean warming contributing to the melting of Arctic sea ice (Dörr et al., 2024) and ice shelves (Slater & Straneo, 2022)? Will deep convection migrate northward, accelerating changes to the cryosphere (Heuzé & Liu, 2024)? Will the Beaufort Gyre collapse and release its stored freshwater (Timmermans & Marshall, 2020)?

Oceanic climate change throughout the ocean basins is addressed by this opportunity. How will western boundary currents and gyres respond to climate change (Sen Gupta et al., 2021)? How will eastern boundary currents and upwelling respond (Bograd et al., 2023)? Globally, the three-dimensional dynamics of the ocean not only control thermosteric and halosteric sea level changes at regional scales (Griffies et al., 2014; Fox Kemper et al., 2021), but also manometric sea level changes in shallow oceans (Samanta et al., 2024; Jevrejeva et al., 2024). These sea level drivers

will affect the future of human activities in coastal regions across the globe.

To achieve these goals, the Ocean Changes, Drivers and Impacts Opportunity requests the datasets necessary to analyze three-dimensional ocean processes and their time evolution as well as their feedbacks on the atmosphere and cryosphere, going beyond those defined as baselines in Juckes et al. (2024). Most of these diagnostics and the corresponding variables were defined and some were introduced in the contribution of OMIP to CMIP6 (Griffies et al.,

2016, Orr et al., 2017). The main variable groups attached to this opportunity are inherited from Griffies et al. (2016), but have here been prioritized differently based on the experience gained from CMIP6 and a desire to reduce the size of the data requested: for example, the scalar fields in table H1 of Griffies et al. (2016) have been split into two variable groups: *omip_scalar_high_priority* and *omip_scalars_low priority*. The extensive list of variables provided by Griffies et al. (2016) covers most of the needs of this opportunity across different types of ocean models, providing continuity

between CMIP6 and CMIP7 as ocean models evolve. Further developments and refinements impacting ocean variables are described in Fox-Kemper et al. (2025, in preparation). A novelty in CMIP7 is the *ocean_mesoscale* variable group, which contains variables essential for analyzing the output of eddying ocean models. In models with a horizontal resolution of ¼° or higher, ocean eddies are no longer entirely parameterized but largely resolved, requiring additional variables for heat and salt transport to accurately compute their contribution to the heat and salt budgets. Given that

mesoscale eddy activity has changed throughout the historical period (Martínez-Moreno et al., 2021) and is projected to continue evolving in the future (Beech et al., 2022), being able to better quantify these changes will be instrumental to more robustly assess climate simulations. Finally, some new variables (in the variable group *int_ocean_budgets*) include vertically-integrated heat and salt content, intended to more easily track the large-scale changes in the energy and freshwater cycles and compare to observations, and thereby to inform the energy budgeting of the whole Earth

system. While these variables can be calculated instantaneously from the three-dimensional temperature, salinity, and



grid specifications, the increasing complexity of the vertical coordinates used in modern models makes this task complex. Furthermore, because of the limits of observing technology, ranges of depth easily accessed by bathythermograph, Argo floats, and other tools have made for a standard set of layers based on hydrostatic pressure ranges in the observational literature (0-300 m, 0-700 m, 0-2000 m and total depth, where meter ranges imply their hydrostatic pressure equivalents). These new variables, calculated online, will facilitate the evaluation of ocean models using in-situ observations (e.g., Eyring et al, 2021).

| Variable group | Reason for inclusion |
|---|---|
| *baseline_monthly* | Monthly mean ocean variables to get an overview of the ocean state, as well as atmosphere and sea ice variables needed to understand drivers of ocean changes. |
| *baseline_fixed* | Basic time invariant information about all components of the coupled model, including key ocean information such as the bathymetry and grid. |
| *ocean_grid* | Essential variables to describe grid areas and volumes. Note that in many ocean models, cell thicknesses and volumes are time-dependent. |
| *ocean_grid_low_priority* | Additional variables needed to better describe the ocean grid (cell lengths and thicknesses corresponding to different variables, temperature, salinity or velocities). This group also includes time dependent cell areas, relevant for some models. |
| *ocean_mesoscale* | Daily variables required to analyze eddying ocean models, for example, daily sea surface height, as well as monthly output of three-dimensional heat fluxes necessary to assess the eddy contribution to the ocean transports of heat and salt. |
| *ocean_meridional_overturning_streamfunctions* | The ocean streamfunctions in density and depth space in each ocean basin describe the large-scale ocean circulations, which are essential for understanding ocean changes and potential drivers of these changes (Griffies et al. 2016, table I1). |
| *omip_budgets* | Variables describing the various contributions to changes of heat and salt in each model grid cell. This group has low priority (Griffies et al. 2016, table L1). |
| *int_ocean_budgets* | Vertically-integrated energy and salt content in layers are necessary for model validation with in-situ ocean observations. |
| *omip_parameterizations* | Variables describing the contribution of parameterizations of lateral mixing |



| | |
|---|---|
| | to ocean budgets of tracers and momentum (Griffies et al. 2016, table N1). |
| *omip_scalars_high_priority* | Scalar fields required for the description of the ocean state (Griffies et al. 2016, table H1). |
| *omip_scalars_low_priority* | Variance of scalar fields at monthly frequency, priority 3 in Griffies et al. 2016, table H1. |
| *omip_transports_high_priority* | Ocean transports of mass, heat, and salt (Griffies et al. 2016, table I1, priority 1). |
| *omip_transports_medium_priority* | Ocean transport variables at priority 2 from Griffies et al. (2016, tables I1 and J1). This includes the mass transports through selected straits, and components of basin scale transports (overturning versus gyre). |
| *omip_transports_low_priority* | Contributions from parameterizations to the overturning streamfunction (Griffies et al. 2016, table I1). |
| *omip_vectors_high_priority* | Three-dimensional fields of ocean velocities are needed to analyze the ocean circulation. |
| *omip_surface_fluxes_high_priority* | Water and heat fluxes at the ocean surface are required to close the heat and water budgets of the ocean. |
| *omip_surface_fluxes_medium_priority* | Individual components of the heat and water fluxes required for process understanding (Griffies et al. 2016, tables K1, K2 and K3). |
| *omip_momentum_fluxes_high_priority* | Momentum fluxes to characterize the wind stress at the ocean surface, which is a key forcing mechanism for the ocean circulation. |

**Table 2:** Ocean Changes, Drivers and Impacts Opportunity variable groups and their rationales for inclusion.

**Sea Ice Changes, Drivers and Impacts Opportunity (ID 73)**

Sea ice plays a role of prime importance in the climate system because of its widespread coverage and very different physical properties compared to the ocean and atmosphere. Often compared to a white blanket, sea ice reflects solar radiation back to space and acts as a thermal insulator by drastically reducing the turbulent heat fluxes from the warmer ocean to the colder atmosphere during winter (Zampieri et al., 2024) and mitigating the warming of the ocean by absorbing heat during the polar summers (Li & Liu, 2022). Sea ice changes because of thermodynamic processes, that

change its mass and heat content, and by so-called dynamic processes, responsible for its drift and deformation. Sea ice is not only a mediator of atmospheric-oceanic heat exchange but also the transfer of momentum from the atmosphere to the ocean (i.e., stresses). The seasonal formation of new sea ice and consequent rejection of salty brine into the ocean reduces the stability of the underlying water column and contributes to forming the world's densest waters. These feed the ocean thermohaline circulation on a global scale (Fox-Kemper et al. 2021). In contrast, the melting of sea ice



freshens the surface ocean and has a stabilizing effect on its vertical stratification (Linders & Björk, 2013). Sea ice is not a continuous rigid plate, but rather composed of a dynamic ensemble of floes in non-uniform motion, with varying thickness and sizes that span over several orders of magnitude (Gherardi & Lagomarsino, 2015), making its accurate representation in climate model simulations challenging.

Changes in sea ice have significant impacts on atmosphere, ocean, marine biogeochemistry, ecosystems, and human

activities. In recent decades, the Arctic sea ice cover has been declining rapidly, and the signal of a forced sea ice retreat has clearly emerged from the background noise of year-to-year variability (Notz & SIMIP Community, 2020). In contrast, the Antarctic sea ice area showed a small positive trend over 1979–2015, but this was followed by sudden decreases in recent years (Eayrs et al., 2021), which suggests that the Antarctic might be rapidly transitioning to a new, low sea ice state (Purich and Doddridge, 2023; Hobbs et al., 2024) because of thermodynamic processes (Himmich et

al., 2024). Given the importance of sea ice in the climate system and the swift transitions currently occurring in the polar regions, it is essential to advance our understanding of past and future sea ice changes, their driving mechanisms, and their impacts.

Ahead of CMIP6, the SIMIP Community developed a protocol detailing a standard for sea ice model output to streamline, and hence simplify, the analysis of the simulated sea ice evolution in model inter-comparison projects (Notz

et al., 2016). This protocol allowed researchers to conduct process-level analysis of the three main budgets that cover the evolution of sea ice, namely the heat, momentum and mass budgets (e.g., Keen et al., 2021; Watts et al., 2021; Zanowski et al., 2021; Lee et al., 2023; Frankignoul et al., 2024; Kuang et al., 2024). Notwithstanding this massive effort and the success it has enabled in sea ice modeling studies, simulations of past and future sea ice evolution from CMIP6 models still exhibit a large inter-model spread and fail at capturing important sensitivities, including the

response of sea ice area to global mean temperature change (Notz & SIMIP Community, 2020; Roach et al., 2020).

To achieve a process-based assessment of the sea ice evolution, we need variables to diagnose the state of Arctic and Antarctic sea ice, understand the mechanisms driving changes, and assess impacts. The Sea Ice Changes, Drivers and Impacts Opportunity seeks to provide the necessary model outputs to reproduce and build on the work done with CMIP6 simulations, leading to a better understanding and, ultimately, reducing biases and errors in simulation of sea

ice. The variable groups were created mainly following Appendices D-H of Notz et al. (2016), which grouped variables based on the following categories: sea ice state variables, tendencies of sea ice mass, heat and freshwater fluxes, sea ice dynamics, and integrated quantities. The prioritization of each variable group was based on past usage/download of the variables included in the group by the SIMIP Community as well as considerations related to the amount of data produced. In addition, the *seaice_gcos_ecv* variable group was created to align with the seven essential climate sea ice

variables defined by the Global Climate Observing System (GCOS; Lavergne et al., 2022)  to facilitate the evaluation of model output against observational products. Finally, a few new variables were added compared to CMIP6, namely the effective melt pond fraction to allow for direct comparison with observations, as well as the integrated mass of



snow on sea ice for each hemisphere to allow for a complete high-level analysis of the total mass of the cryosphere in the Earth system (see Annex 2 for more detail).


| Variable group | Reason for inclusion |
|---|---|
| *baseline_monthly* | Basic atmosphere and ocean variables necessary to understand the coupled processes driving sea ice changes. |
| *seaice_budget_mass_monthly* | Monthly sea ice variables necessary to analyze the evolution of the sea ice mass budget and quantify the physical origin and location of sea ice growth and melt. |
| *seaice_budget_area_monthly* | Monthly sea ice variables necessary to analyze the evolution of the sea ice area budget and quantify the physical origin and location of sea ice area changes. |
| *seaice_budget_energy_monthly* | Monthly sea ice variables necessary to analyze the evolution of the sea ice energy budget and understand the drivers of sea ice mass tendencies, with the different fluxes requested over the sea ice covered portion of the grid cell. |
| *seaice_budget_freshwater_monthly* | Monthly sea ice variables necessary to analyze the contribution of sea ice to the ocean freshwater budget and understand the interaction of sea ice with the hydrological cycle of the Earth, specifically the storage of both salt and freshwater from sea ice growth or melt. |
| *seaice_state_monthly_basic* | Monthly sea ice variables needed for assessing the seasonal cycle and long-term evolution of the sea ice state. |
| *seaice_state_monthly_advanced* | Monthly sea ice variables necessary to allow advanced process understanding of sea ice, its spatial distribution and temporal evolution, beyond what is included in the *seaice_state_monthly_basic* variable group. |
| *seaice_state_daily_basic* | Daily sea ice variables necessary to analyze the evolution of sea ice characteristics at the sub-seasonal scale. |
| *seaice_global_monthly_basic* | Hemispheric-integrated measures of monthly sea ice area, extent, volume and snow mass used to examine the large-scale sea ice evolution. |
| *seaice_global_monthly_advanced* | Net sea ice mass transport through the four gates of the Arctic Ocean (Fram Strait, Canadian Arctic Archipelago, Barents Sea Opening, Bering Strait) used to examine changes in export of Arctic sea ice (not relevant to Antarctic sea ice). |
| *seaice_global_daily_basic* | Hemispheric-integrated measures of daily sea ice area, extent, volume, and snow mass needed to assess changes in seasonality (e.g., the day of year when a given volume of sea ice is passed). |



| | |
|---|---|
| *seaice_dynamics_basic* | Sea ice variables needed for basic assessment of sea ice dynamics (drift and deformation processes). |
| *seaice_dynamics_advanced* | Sea ice variables needed for a more detailed assessment of sea ice dynamics, including horizontal sea ice and snow mass transport terms, contributors to the sea ice horizontal momentum budget, and invariants of the stress and strain rate tensors. |
| *seaice_gcos_ecv* | Sea ice variables defined as Essential Climate Variables (ECVs) for the Global Climate Observing System (GCOS, Lavergne et al., 2022) to consistently evaluate model output against observational products. |

**Table 3:** Sea Ice Changes, Drivers and Impacts Opportunity variable groups and their rationales for inclusion.

**Paleoclimate Research at the Interface between Past, Present, and Future Opportunity (ID 51)**

In paleoclimate research, the integration of modeling (e.g., Sherriff-Tadano and Klockmann, 2021) and model-independent data from geologic or glaciologic archives (e.g., Gulev et al., 2021, Fox-Kemper et al., 2021a) is possible,

providing an opportunity for testing the skill and robustness of climate models and improving confidence in projections of future climate change (e.g., Masson-Delmotte et al., 2013; Kageyama et al., 2024; Haywood et al., 2019). The study of paleoclimates provides a critical window into climate conditions that differ significantly from the present, including periods with higher global temperatures and different atmospheric $CO_2$ concentrations. For example, about 3 million years ago, a much warmer than present Arctic supported large herbivores at atmospheric $CO_2$ levels comparable to

today (Rybczynski et al., 2013; de La Vega et al., 2020). Such paleoclimate reconstructions offer verification data, provide a test for climate model performance under warm climates bearing similarity to future projections, and highlight the exceptional nature of current anthropogenic climate change.

The goal of this opportunity is to leverage paleoclimate records and experiments to evaluate model performance across a broad range of climate states beyond the instrumental record and to reflect on thresholds in the Earth system under

conditions that are extremely different from today. This includes quantifying uncertainties and model-discord (e.g., Kageyama et al., 2021) and identifying potential biases in model simulations, particularly for extreme or non-analog conditions. Paleoclimate simulations allow researchers to assess the capability of models to reproduce reconstructed climatic features such as polar amplification, flat meridional temperature gradients, or past sea ice extent, which are often poorly captured in climate models (Dowsett et al., 2013). By integrating geological and glaciological evidence

with climate model outputs, model skill can be evaluated for climate states that are outside the range of modern observational conditions where models are developed and calibrated to succeed (e.g., Zhu et al., 2020). Overconfidence in model parameter calibrations can be reduced (e.g., Lohmann et al., 2022), leading to better constraints on feedback, tipping points, and the dynamics of large-scale circulation systems (e.g., Armstrong McKay et al., 2022; Wunderling et





al., 2024; Brown et al., 2020; Cooper et al., 2024). The inclusion of paleoclimate benchmarks helps evaluate the

stability and reliability of climate model components under diverse forcing regimes, ultimately informing future

scenario projections.

To achieve these aims, this opportunity incorporates a broad suite of CMIP7 AFT experiments, not just the abrupt-127k

simulation, so that other PMIP7 simulations of climates from distant past (e.g., Last Interglacial, mid-Holocene,

Pliocene) may be compared with historical and future period simulations.  The variable request includes diagnostics of

the atmosphere, ocean, land surface, and sea ice to support model-data comparisons and analysis of key climate

processes across timescales. Sea ice variables, many aligned with SIMIP (Notz et al., 2016), are included to examine

cryosphere evolution under past, present and future warm climates. In addition, specific variables support paleoclimate-

focused research areas such as stable water isotopes, data assimilation, and coupled carbon-cycle feedback. These

outputs will enable the evaluation of model performance across a wide range of climatic states, improve understanding

of processes driving climate variability and change, and enhance integration between diverse research communities

within CMIP7 that focus on past or future climates.

| Variable group | Reason for inclusion |
|---|---|
| *paleo_fx* | Additional metrics to characterize paleo-geographies, such as different land, ocean, and lake distributions, so these changes can be taken into account. |
| *paleo_atmosphere* | Selected three-dimensional atmospheric quantities to quantify the state of the paleoclimatic atmosphere, illustrating, for example, differences in large-scale transport regimes and modes of internal variability. |
| *paleo_radiation_fluxes* | Variables to characterize heat and radiation fluxes and the energy balance of the Earth system across time scales. |
| *paleo_land_atmosphere_surface* | Variables used in paleoclimate studies for characterization of land and atmosphere and their interactions, addressing broad aspects like temperature, precipitation, and the cycling of energy and water, that may be very different from today. Paleoweather extremes are included via selected daily mean variables. |
| *paleo_permafrost* | Variables for the study of permafrost and for driving offline permafrost models. |
| *paleo_ocean* | Variables for paleoclimate research that characterize conditions and fluxes at the ocean surface, that quantify links between surface ocean and deep ocean, and that provide integrals of global salt and ocean temperature inventories. They support monitoring the progress of model equilibration and highlight differences |



| | in large-scale patterns between different climate states. |
|---|---|
| *paleo_ocean_3D* | Three-dimensional ocean quantities that enable studying shifts in heat and salt between basins, latitudes, and across the water column, and help track related large-scale changes in ocean circulation commonly found in paleoclimate. |
| *paleo_ocean_transports* | Variables to characterize past ocean transport regimes that may differ substantially from today. |
| *paleo_stable_isotopes* | Variables needed to characterize the hydrological cycle across time-scales and that enable direct comparison of models and stable-isotope-based proxy records. |
| *paleo_cryosphere_high_priority* | High-priority variables to study the state of the cryosphere (sea ice in particular), as well as fluxes and transports that are key for cryosphere dynamics and may substantially differ from today. These include variables that are necessary to compute the albedo of different Earth system components. |
| *paleo_cryosphere_medium_priority* | Variables (exclusively from SImon) that allow a closer look at the state of sea ice and at the dynamics that drive sea ice evolution. |
| *paleo_cryosphere_low_priority* | Variables related to sea ice melt and growth processes, and to sea ice transport, that are less commonly analyzed, but that help understand further details of sea ice dynamics. |
| *paleodata_assimilation* | Variables needed for exploring deviations between modeled and recorded past climate. |

**Table 4:** Paleoclimate Research at the Interface between Past, Present, and Future Opportunity variable groups and their rationales for inclusion.

**Causality of Polar Amplification Opportunity (ID 13)**

Polar amplification—enhanced warming at high latitudes relative to global mean temperature warming—is a robust feature of global climate change identified more than a century ago (Arrhenius, 1896). While polar amplification occurs in remote regions of the planet, it has global consequences. High-latitude climate change affects sea level rise, ocean and atmospheric circulation patterns, and the carbon cycle in addition to the local impacts on ecosystems and human systems (e.g., Constable et al., 2022). The rate of Arctic climate change has implications for economic development in the region, natural resource exploration, geopolitics, and adaptation (Nanni et al., 2024). However, the uncertainty in high-latitude climate projections continues to be greater than in other regions of the globe across CMIP generations (Holland and Bitz, 2003; Hahn et al., 2021). The Causality of Polar Amplification Opportunity seeks to assess the processes driving polar amplification and enable the determination of the contributions of high-frequency processes to the inter-model spread.



At the heart of polar amplification uncertainty is the current inability to attribute causality to the polar climate changes seen in observations and simulated by models. While polar amplification is a coupled atmosphere-sea ice-ocean process, a robust, quantitative understanding of the causal factors remains unclear (Manabe & Stouffer, 1980; Previdi et al., 2021; Taylor et al., 2022). Evidence suggests that the processes and feedback between sea ice, atmosphere, and ocean that are central to polar amplification unfold at high frequencies (e.g., from days to weeks). For instance, atmospheric rivers have been shown to be responsible for up to 90% of the poleward atmospheric moist energy transport (Newman et al., 2012), and an increase in the frequency of such events penetrating the Arctic accounts for much of the sea ice decline in the Barents-Kara Sea and central Arctic during the ice growth season (Zhang et al., 2023). Such high-frequency variability and the associated interactions with the sea ice pack, atmospheric state, and ocean are posited to be central to the causality of polar amplification (Taylor et al., 2022; Parker et al., 2022; Cardinale & Rose, 2023).

Contributions from high-frequency variability (e.g., atmospheric rivers and cyclones) to the inter-model spread in polar amplification cannot be accurately assessed with monthly mean model outputs. In the absence of sub-monthly data, the atmospheric energy transport by transient eddies can only be calculated as the residual between the total atmospheric energy transport and the transport by the mean meridional circulation (e.g., Donohoe et al., 2020). Furthermore, there is increasing recognition that the character (moist versus dry, e.g., Graversen & Burtu, 2016) and vertical structure (e.g., Cardinale & Rose, 2023) of the atmospheric energy transport matter more than the total amount. To enable studies that advance our understanding of the causal mechanisms of polar amplification, this opportunity contains the necessary model outputs from sea ice, atmosphere, and ocean components to assess the contributions of high-frequency processes to polar amplification and the inter-model spread in projections of polar climate change. Key sea ice variables include thickness, concentration, and surface energy budget diagnostics. Atmospheric variables such as wind, temperature, specific humidity, and geopotential height at all model levels and surface pressure are needed (Cox et al., 2024). For the ocean, mixed layer depth and thermal energy transport are most useful in studying polar amplification.

| Variable group | Reason for inclusion |
|---|---|
| *baseline_daily* | Daily atmospheric temperature, humidity, and wind profiles and radiative fluxes necessary to enable the characterization of the high-frequency cyclones and atmospheric rivers and their structure, and to analyze the influence on the sea ice pack and the inter-model differences. |
| *seaice_state_daily_basic* | Daily sea ice concentration and thickness necessary to analyze the sea ice response to high-frequency atmospheric and ocean variability. |
| *seaice_state_daily_advanced* | Daily advanced sea ice variables necessary to analyze the high-frequency evolution of sea ice properties and the response to atmospheric and ocean variability. |




| *seaice_budget_energy_daily* | Daily sea ice variables necessary to analyze the evolution of the sea ice energy budget and understand the drivers of high-frequency sea ice mass tendencies, with the different fluxes requested over the sea ice covered portion of the grid cell. |
|---|---|
| *atmospheric_transports* | Daily vertically-integrated horizontal transport of dry static energy and moisture necessary to identify anomalous transport events and analyze the interactions with the sea ice pack and ocean. |
| *ocean_mesoscale* | Oceanic mesoscale variables and ocean heat transport necessary to assess the interactions between ocean variability and sea ice property evolution. |

**Table 5:** Causality of Polar Amplification Opportunity variable groups and their rationales for inclusion.

**Ocean Extremes Opportunity (ID 49)**

Ocean extreme events are by definition uncommon and intense occurrences, often impacting marine life and coastal environments. Marine heatwaves are an example of short-term extreme oceanic events. These rare events occupy the tail of the upper ocean temperature distribution—typically defined as being in the 90th percentile of the climatology or
a similar threshold—and can persist for days to months (Hobday et al., 2018; Oliver et al., 2021). They primarily affect the mixed layer which overlaps with the euphotic zone where most ocean photosynthesis occurs (Smith et al., 2024; Smale et al., 2019), and they have a large impact on reef-forming corals that provide critical ecosystem services such as coastal defense, subsistence fisheries, and nursery habitats for commercially important fish and shellfish (Gomes et al., 2024).   Other ocean extremes, including anomalous subsurface oxygen or pH, salinity changes, or extreme sea level
events, can occur independently or as compound events with marine heatwaves (Gruber et al. 2021; Burger et al., 2022; Ren & Rudnick, 2021; Han et al., 2022). Mesoscale eddies, fronts, or other anomalies in surface velocity or vorticity are often associated with these compound events. The Ocean Extremes Opportunity aims to investigate and address the impacts of extreme ocean conditions such as marine heatwaves, hypoxic zones, extreme salinities, sea level extremes and storm surges, ocean acidification, and compound events.

As the climate changes, extreme conditions are becoming more frequent and severe in many regions, posing significant risks to marine ecosystems, livelihoods (e.g., fisheries), and coastal communities and infrastructure (Gruber et al., 2021; Fox-Kemper et al., 2021a; van de Wal et al., 2024; Smith et al., 2025). Studying these characteristics requires high-frequency surface data, built up over whole scenario time series to establish climatological ranges and capture events and changing likelihoods. Coastal hazards in the form of extreme sea levels cause billions of dollars of
damage globally, and are projected to increase in frequency (Fox-Kemper et al. 2021a). Extreme sea levels are caused by the complex interplay of multiple contributors, including astronomical tides, storm surges, waves, and sea level variability (Idier et al., 2019; Melet et al., 2024), which vary on sub-daily to interannual frequencies, as well as climate change trends. Storm surges and extreme waves are caused by prevailing atmospheric surface pressure and wind



conditions. Their magnitude and the extent of their impact on a coastline are influenced by bathymetry, coastal
morphology, tidal amplitude, and tidal cycle, and are further exacerbated by sea level rise (Bernier et al., 2024).
Additionally, coincident pluvial and fluvial flooding can compound the severity of inundation and erosion, especially
during extreme weather events. These factors highlight the need for comprehensive climate data, including winds,
atmospheric pressure, precipitation, and variables associated with ocean circulation (currents, temperature, salinity), as
well as other environmental variables to better understand current and future coastal hazards risk. Long-term
assessment of ocean extremes is important for identifying vulnerable regions, assessing infrastructure and ecosystem
viability, designing coastal protection to appropriate levels, and enabling various adaptation measures to be considered
and tested via sensitivity testing. In the coming decades, ocean extremes and associated floods and erosion are likely to
remain a leading cause of natural disasters because of the increasing frequency and intensity of extremes combined
with increased coastal development associated with greater exposure (Bernier et al., 2024).
To better understand the transient nature of ocean extreme events, the Ocean Extremes Opportunity seeks to capture the
statistics of these events and how they compare to observed statistics during the historical period, their changing
likelihoods, and their sensitivity to anthropogenic forcing through a wide range of scenarios. Studying the
characteristics of ocean extremes requires high-frequency surface data, built up over whole scenario time series to
establish climatological ranges and capture events and changing likelihoods. Furthermore, high-frequency subsurface
marine heatwave events are also under study and have been found to be more frequent under climate change (Sun et al.,
2023). By including surface values of temperature, salinity, velocity, sea level, pH and a minimal amount of subsurface
information (200 m depth only) including also oxygen concentration, this opportunity will allow for a better
understanding of the mechanisms and stratification conditions associated with these extremes. Additionally, variables
needed to quantify storm surges, which exacerbate extremes, are included. These data will be useful in understanding
such events, but they will also enable the study of their correlation with modes of climate variability such as ENSO,
with changes in ocean currents and stratification, as well as the evaluation of potential impacts on vulnerable
ecosystems and coastal regions. Surface fluxes, which are part of many other opportunities, can also be used for causal
inference about specific events.

| Variable group | Reason for inclusion |
| --- | --- |
| *ocean_acidification_oxygen_extremes* | Variables needed to capture acidification and low oxygen extreme events and related compound events. |
| *ocean_KE_vorticity_extremes* | Variables needed to capture transport anomalies, eddies, and similar phenomena. |
| *ocean_temperature_extremes* | Variables needed to capture marine heatwaves and related compound events. |
| *sea_level_extremes* | Variables needed to improve understanding of ocean extremes, coastal |





| | |
|---|---|
| | inundation and erosion, community and ecosystem vulnerability and response. |
| *mixed_layer_extremes* | Variables needed to indicate the rapid evolution of the upper ocean, the rate at which it will saturate in absorbing heat, carbon, and oxygen, and the structures that support extreme events. |
| *surgemip_variables* | Variables needed to improve understanding of ocean extremes, coastal inundation and erosion, community and ecosystem vulnerability and response using CMIP7 directly, in downstream/offline tools, or to force high-resolution regional ocean-wave models, etc. |

**Table 6:** Ocean Extremes Opportunity variable groups and their rationales for inclusion.

**Advancing Wind Wave Climate Modeling for Coastal Zone Dynamics, Impacts, and Risk Assessment Opportunity (ID 24)**

Wind waves are ocean surface gravity waves generated by the action of the wind blowing across the ocean surface over a certain distance known as fetch (Young, 1999; Holthuijsen et al., 2007). Understanding how wind wave climate

evolves on global and regional scales is essential for predicting coastal hazards, erosion, and other wave-related impacts as well as for supporting marine operations and climate adaptation strategies such as renewable energy activities (Casas-Prat et al., 2024). However, despite the many role surface waves play in the coupled climate system (Cavaleri et al., 2012), studies on wind wave climate projections are often hindered by significant uncertainties (Morim et al., 2019), particularly at the extremes (Meucci et al., 2020) which are of the utmost importance for the safety of

offshore and coastal activities. Currently, most global climate models participating in the CMIP effort do not include an active wave component (Casas-Prat et al., 2024). To enhance our understanding of wind wave climate and to improve future projections, high-resolution data on ocean surface wind speed and sea ice concentration are crucial to drive offline wave simulations. The purpose of this opportunity is to enable high-resolution and flexible modeling of ocean surface wind wave climates, independent of ESM outputs, in support of global and regional risk and impact

assessments, building on the internationally coordinated effort of the Coordinated Ocean Wave Climate Project (COWCliP). By offering computational efficiency and  spatial detail, this method provides actionable data for stakeholders involved in climate adaptation and marine planning.

The societal benefits of improved wind wave modeling extend far beyond academia. Coastal hazards driven by wind waves pose significant risks to coastal communities, economies, and ecosystems. Currently, around 15% of the global

population lives within 10 km of the coast, which equates to more than a billion people (Cosby et al. 2024). In terms of economic risk, coastal areas are home to major cities, critical infrastructure, and industries like shipping, fisheries, and tourism. Coastal cities and agglomerations have increased in number by 4.5 times since 1945 (Barragán et al., 2015) and the population living in proximity to coasts is projected to continue to increase, further intensifying the



vulnerability of coastal areas to hazards such as storm surge, flooding, and erosion. As climate change intensifies, the

frequency of extreme wave events is likely to rise (Meucci et al., 2020; Lobeto et al., 2021; O'Grady et al., 2021), putting more lives and assets in danger.

Furthermore, the growing demand for sustainable energy sources highlights a critical need for comprehensive assessments of wave energy, offshore wind energy, and other types of renewable energy that can be deployed on or near coastlines. These assessments depend on the potential of high-resolution coastal wind wave information, which is

fundamental to evaluating the feasibility of wave and offshore wind energy as renewable resources (Kulkarni et al., 2018; Jung et al., 2024). Such data help identify optimal locations for wave and wind energy farms by analyzing long-term wave climate patterns and understanding climate variations. Large-scale atmospheric and oceanic variability, such as the ENSO and the North Atlantic Oscillation (NAO), climactic trends, and connected regional, coastal wind and wave variability at high resolution are all important for long-term planning (Tseng et al., 2024). The harsh marine

environment poses strong currents, high waves, and corrosive conditions, which can impact the safety, longevity and reliability of infrastructure. Detailed risk assessments help identify potential hazards, inform mitigation strategies, and ensure compliance with international safety standards, safeguarding both workers and the environment.

For effective offline wind wave climate modeling, high-resolution temporal and spatial data are essential. Wind patterns can vary significantly over short time scales and distances, and the interaction between wind, waves, and ice

requires detailed data to capture these dynamics. Similarly, sea ice concentration data is critical for understanding how ice attenuates or reflects wave energy, as well as how a changing ice cover due to warming temperatures alters wave patterns. Without high-resolution data, models may miss localized phenomena such as extreme wave events or changes in coastal wave energy distribution, which are crucial for understanding and predicting coastal erosion and other hazards. By incorporating finer-scale data into wave models, we can better understand the spatial variability of wave

energy, identify vulnerable coastal zones, and assess the future risks posed by changing wave climates.

| Variable group | Reason for inclusion |
|---|---|
| *baseline_monthly* | Variables needed to improve understanding of average wind wave climate patterns. |
| *baseline_daily* | Variables needed to understand wave responses to extreme events and risk calculations. |
| *baseline_subdaily* | Variables to use as forcing field for global and regional spectral wave climate modeling. |
| *seaice_state_daily_basic* | Variables that are essential for accurately representing ice-induced wave attenuation and interactions, providing a crucial forcing field for global wind wave climate models. |
| *cowclip_wind_wave_varia* | Variables to improve understanding of wind wave extremes climate crucial for |





| *bles* | future coastal safety and adaptation measures. |
|---|---|

**Table 7:** Advancing Wind Wave Climate Modeling for Coastal Zone Dynamics, Impacts, and Risk Assessment Opportunity variable groups and their rationales for inclusion.

**Effects and Feedbacks of Wind-Driven Ocean Surface Waves Coupled Within Earth System Models**
**Opportunity (ID 68)**

Traditionally, ocean surface wind wave climate studies, such as those highlighted in the previous opportunity (ID 24), rely on high-frequency output from ESMs for offline execution of a comprehensive suite of statistically or dynamically downscaled wave climate ensembles for past and future conditions (e.g., Hemer et al., 2013; Morim et al., 2019; Meucci et al., 2020, 2024; Casas-Prat et al., 2024). However, numerous studies have demonstrated that ocean waves
play a critical role in regulating Earth's climate system by influencing the exchange of energy, momentum, and mass between the ocean and atmosphere (e.g., Babanin, 2006; Belcher et al., 2012; Cavaleri et al., 2012; Qiao et al., 2013, 2016; Li et al., 2016; Li & Fox-Kemper, 2017; Li et al., 2019; Fox-Kemper et al., 2021b). Some of these feedback mechanisms have been recognized by meteorological institutions, where two-way wave-atmosphere interactions are now incorporated into weather prediction models (e.g., Janssen, 1991; ECMWF, 2024). Ongoing research seeks to
clarify to what extent ocean surface waves influence the climate on longer timescales. The goal of this opportunity is to support research on the influence of ocean surface waves on the climate system, including their feedbacks within ESMs and their long-term role as climate drivers. Unlike ID 24, this is not part of an established coordinated modeling effort, but rather aims to advance understanding and foster new developments in surface waves that are coupled online as a component within ESMs.

Breaking waves, for instance, contribute to the generation of sea spray, which has significant implications for cloud formation (Veron, 2015; DeMott et al., 2016; Brumer et al., 2017; Deike et al., 2022). Additionally, wave-driven processes modulate air-sea gas exchange, particularly affecting $CO_2$ uptake (Deike & Melville, 2018; Woolf et al., 2019). Ocean waves play a significant role not only in the modulation of the atmospheric boundary layer, but also influence ocean mixed layer depths through both breaking and non-breaking motions that induce turbulence via
Langmuir instabilities or other mechanisms (Qiao et al., 2013, 2016; Li et al., 2016; Li & Fox-Kemper, 2017; Li et al., 2019). However, fully-coupled interactions between waves, mean flow, and turbulence still have many unknowns (Kantha & Clayson, 2004; Suzuki et al., 2016; Wu et al., 2019; Fox-Kemper et al., 2021b).

In the polar regions, ocean waves play a crucial role in shaping sea ice dynamics, particularly in the marginal ice zone (e.g., Roach et al., 2018). As waves propagate through this region, they can fracture the ice, breaking it into smaller
floes. This process enhances both lateral melting (melting from the edges of ice floes) and basal melting (melting from beneath the ice) by increasing the ice's exposure to warmer ocean waters (Alberello et al., 2022). Additionally, waves alter air-sea-ice flux exchanges, influencing how heat, moisture, and momentum are transferred between the ocean and the atmosphere (Bennetts et al., 2024). Furthermore, the reduction of sea ice due to storm-driven wave activity can have



far-reaching consequences (Kohout et al., 2014; Blanchard-Wrigglesworth, 2024). When sea ice is absent or
significantly weakened, ice shelves become more exposed to powerful ocean swells, leading to ice shelf disintegration
and an acceleration of the loss of polar ice masses (Massom et al., 2018).

Given the increasing recognition of ocean waves as key components of the Earth system, several modeling centers have
started incorporating wave coupling into ESMs and Global Climate Models (GCMs; Qiao et al., 2013, 2016; Li et al.,
2016, 2017; Reichl & Li, 2019; Bao et al., 2020; Danabasoglu et al., 2020; Brus et al., 2021). This coupling should
allow for a more accurate representation of air-sea interactions, improving the simulation of weather and climate
phenomena (Fox-Kemper et al., 2019; Fox-Kemper et al., 2021b). By integrating wave processes into climate models,
researchers aim to enhance predictions of extreme events and long-term climate variability as well as to better
understand the roles of waves in the climate system and the feedback processes they entail.

High temporal and spatial resolution outputs of wave parameters from coupled climate models, including significant
wave height, mean and peak wave periods, and mean wave direction, are particularly valuable for comparing wave
climate simulations performed with the traditional stand-alone spectral wave models, which rely solely on atmospheric
forcing. By analyzing wave fields from coupled climate models alongside conventional wave climate model outputs,
we can start assessing the added value of wave coupling in representing long-term wave climate variability and extreme
wave events. In addition, this opportunity includes some key ocean and sea ice variables so that the impacts of
parameterizations and wave effects mentioned above can be assessed. Since surface waves affects multiple aspects of
the climate system, this opportunity has the potential to improve the results and findings from other opportunities (e.g.,
ID 13, ID 24, ID 49 and ID 73). We request that modeling centers consider the inclusion of these wave variables in
their coupled simulation outputs and make them accessible for model intercomparison research. This will enable more
comprehensive evaluations of wave climate projections, leading to an improved understanding of wave-driven
feedback and better-informed applications in climate science, coastal hazard assessment, and marine operations.

| Variable group | Reason for inclusion |
|---|---|
| *baseline_monthly* | Variables that capture the baseline behavior of the model. |
| *baseline_daily* | Variables that capture key high-frequency behaviors that may be connected to surface wave feedback. |
| *baseline_subdaily* | Variables that capture key high-frequency behaviors that may be connected to surface wave growth and feedback. |
| *baseline_fixed* | Variables that capture the baseline behavior of the model. |
| *sfc_waves* | Wave variables that capture the essential online statistics of surface waves so that they may be related to winds, currents, and other model behavior. |
| *seaice_state_daily_basic* | Sea ice variables that capture any sea ice-wave coupled dynamics and feedback, |



| | such as wave fracture of floes. |
|---|---|
| *mixed_layer_extremes* | Variables that capture upper ocean coupled dynamics and feedback through parameterizations, such as Langmuir mixing and non-breaking wave-induced turbulence. |
| *ocean_temperature_extremes* | Variables that capture temperature responses to, and covariations with, upper ocean sea ice-wave and sea ice-upper ocean coupled feedback. |

**Table 8:** Effects and Feedbacks of Wind-Driven Ocean Surface Waves Coupled Within Earth System Models Opportunity variable groups and their rationales for inclusion.

**Rapid Evaluation Framework Opportunity (REF) (ID 55)**

The CMIP Rapid Evaluation Framework (Hoffman et al., 2025) was created to evaluate and benchmark the newly available CMIP7 AFT simulations as soon as they are uploaded to the Earth System Grid Federation (ESGF), providing metrics and diagnostics that are available through different open-source evaluation and benchmarking tools. This opportunity contains the set of variables that are needed for the planned diagnostics and metrics for the REF (CMIP Model Benchmarking Task Team, 2024). The selected metrics and diagnostics for the REF to be available for all

CMIP7 AFT experiments were intentionally chosen for very basic evaluations and are not expected to require highly specific variables. The exact selection of variables was also made consistent with the model evaluation diagnostics in Chapter 3 of the latest IPCC report (Eyring et al., 2021). Due to the fixed timeline for the CMIP7 AFT simulations, there is only a short period for the technical implementation of the REF, and therefore the available metrics and diagnostics in this first version of the REF will be limited to a temporal resolution of monthly mean data and about five

metrics/diagnostics per realm. Implementation will be based on a selection made by the community. The realms were chosen specifically to be consistent with the realms used for the data request. Find more information about the REF Opportunity in Dingley et al. (2025).

| Variable group | Reason for inclusion |
|---|---|
| *ref_ocean_and_seaice* | This is the set of variables that is needed for the planned ocean and sea ice diagnostics and metrics of the Rapid Evaluation Framework Opportunity. |

**Table 9:** Rapid Evaluation Framework Opportunity variable groups and their rationales for inclusion.

## 4 Discussion

### 4.1 Prioritization process

The prioritization process, both in terms of selecting, combining, and sometimes rejecting opportunities, and in terms of prioritizing among variables and variable groups, is inevitably imperfect. While the team met frequently, discussed the selected opportunities in detail, and then reviewed the variables collectively and individually for errors or oversights,



there are far too many variables involved to keep a clear perspective on all that is required and most urgent. Building

this team from a group of scientists, few of whom had previously worked together, required effort, as did familiarizing oneself with the procedures, protocols, and software selected by the IPO and CMIP Data Request Task Team for the development of the CMIP7 data request. Added complexity and challenges stemmed from the need to coordinate with other author teams in the data request development process, and doing so strictly remotely. The data request surely would have benefited from a slower, longer process, where opportunities to meet in person, such as at large

conferences, could have been taken. On the other hand, the design of the CMIP7 data request had to be implemented within the tight schedule of the overall CMIP7 AFT and IPCC cycle.

While consistency with the CMIP6 data request is insufficient to ensure a successful CMIP7 outcome, it is at least reassuring that the groundwork laid out in CMIP6 was maintained, particularly in the two "Drivers and Impacts" opportunities (ID 47 & 73). Many of the other opportunities were designed by reference to specific single-model

studies that were successful, and thus the multi-model intercomparisons enabled by this data request have a stable foundation.

## 4.2 Outstanding gaps in ocean and sea ice Earth system processes

Despite the large number of variables, many increases in output frequency over CMIP6, and the inevitably vast quantity of data that this request will trigger to be captured, there are still identifiable gaps in the resulting data request that will

make certain processes remain in the shadows, at least in a multi-model sense. Sub-daily data remains extremely rare, but in many of the processes highlighted in these opportunities (e.g., those involving surface waves or processes that depend on particular phases of the diurnal cycle), that frequency is requested for study. While emphasis on extreme events is mostly on ones that persist over a matter of days, such as heat waves, their peak intensity is of even shorter duration and potentially more impactful. Many of these high-frequency variables are only requested for ocean surface

or near-surface levels, which reduces their data volume substantially. One issue that was not resolved in this data request is how to describe the ocean surface as a coordinate designation—a depth of 0 m is not wholly accurate, as many modeling centers provide the uppermost gridcell average, which is not centered on 0 m. However, no consensus on an improved designation was found in time for this data request. Ocean tides are rarely included in ESMs, but in some prototype simulation they have shown significant climate impacts that persist in averages (Arbic, 2022); in most

of the world a semi-diurnal tide dominates, and thus, sub-daily output is needed for tidal coastal dynamics (e.g., Ahmed et al. 2025).

Although this data request also applies to the models in the HighResMIP simulations, and appreciating that some CMIP models will be fairly high-resolution for all applications, reproduction of real-world spatial detail and heterogeneity remains a challenge for representation and study of many Earth system processes that observations reveal to be

important (Hewitt et al., 2020, 2022). While parameterizations can be made to capture some aspects of these unresolved



or poorly-resolved phenomena on the Earth system, a one-size-fits-all data request, such as that produced for CMIP7, cannot meaningfully capture the variety of parameterization mechanisms or their diagnostics so as to compare them across models. This means that it remains painful to carry out careful multi-model parameterization and high-resolution intercomparison studies, although with diligence and focus they still occur (e.g., Chassignet et al., 2020, Uchida et al.,

2022; Li et al., 2019).

### 4.3 Key reflections from the data request process

In a world of increasing automation and artificial intelligence, the process of constructing these data requests was still frustratingly manual and required human intelligence. One simpler idea would be just to request the most popular variables from the CMIP6 request. While download statistics about previous generations of CMIP models were

collected by the Earth System Grid Federation (which maintains and curates the repository), key flaws in collected statistics limit their utility. Not every download was used; some downloads—especially of large-output variables— were downloaded much less often than they were used (i.e., they were shared among colleagues at modeling centers after being downloaded only once), and many routine analyses (e.g., ocean heat content using multiple equations of state, McDougall et al., 2021) ended up requiring the download of much more data than necessary to obtain key data

subsets. So, while our team did consult these download statistics in prioritizing and confirming key variables, CMIP7 AFT missed a chance to profoundly improve the data request, analysis tools, and data access, while simultaneously reducing data storage requirements. The goal is to make the most of CMIP data worldwide, but that is a lofty aspiration. With added foresight, the data request process could have been mostly automated, and then our team could have focused more on facilitating new scientific insights instead of onerous cross-checking and debugging.

Another aspect of automation and artificial intelligence is the growing use of emulators and machine learning parameterizations. The data requested here are not intentionally suited to such purposes, except in that they are the variables for processes with impacts and effects that are important. Their importance suggests that they are likely to be included in machine learning approaches (e.g., ID 22 & 80 of the Impacts and Adaptation data request; Ruane et al., 2025). However, in fields outside of climate science, many recent successes in machine learning have resulted from

carefully curated benchmark training datasets (e.g., Wu et al., 2018; Hu et al., 2020). Although there are limits to how much a benchmark-trained system applies to the real world (Raji et al., 2021), at present, the bigger issue for climate science is the lack of benchmark climate datasets designed specifically for machine learning applications.

Inevitably, future model generations will involve more data, more complex data structures (e.g., unstructured grid meshes, parameters describing hybrid ML-dynamical models), and more and new questions about how the Earth works

and what we can do to conserve and protect its bounty. The length of assessment reports, the number of papers cited in the assessments, the size of data repositories, and the number of variables requested have all been monotonically increasing. The CMIP enterprise has become central to the study of the Earth system, but if care is not taken to seek out



unifying ideas, streamlined processes, and opportunities to take advantage of automation, it may crumble under its ponderous weight (Stevens, 2024). Sharing our climate science challenges through collaborations with data scientists
and computer scientists is one pathway to lightening the burden, but this data request did not fully accomplish a handshake with those communities through benchmark data.

Finally, the essence of science is prediction of what may occur. Scrutinizing such predictions in the light of observations and experiments is how they are evaluated and improved. The continual race to higher spatial and temporal resolution in climate model output is paired with observations that are increasingly high-resolution and
sophisticated. This data request design process involved mostly scientists who have experience in evaluating models versus observations rather than in collecting observations. We foresee that there is a potential to evaluate models in a much more intricate manner than just decimating and interpolating observations collected by other scientists onto our model grid and then calculating simple biases and errors, as it has often been done in the past. To this end, we suggest that future data request designs could benefit from increased involvement of observationalists. It is a life's work to
become an expert in the limits and advantages of particular observations. Their suggestions during the data request design would have offered new insights.

## 5 Conclusions

The CMIP7 data request for ocean and sea ice variables represents a significant step forward in addressing critical gaps in Earth system modeling. Seven opportunities were selected, covering past, present and future changes in ocean and
polar climate, their drivers, and impacts. By refining variable selection and prioritizing key processes, this effort aims to enhance the representation of oceanic and cryospheric dynamics, ultimately improving climate projections. The inclusion of new variables related to surface waves and extremes underscores the evolving needs of the scientific community. These improvements will not only benefit model intercomparison studies but also provide societal and economic benefits, for example through adaptation to coastal hazards.
Looking ahead, continued collaboration and refinement of the CMIP7 data request will be essential to ensure that model output meets the needs of a wide range of stakeholders, from climate scientists to policymakers. Addressing outstanding gaps, such as high-resolution process representation and improved coupling between ocean, ice, and atmosphere components in coupled ESMs, remains a priority. As models become more sophisticated, and observational constraints improve, CMIP7 will play a pivotal role in advancing our understanding of climate variability and long-
term change. The success of this initiative will ultimately depend on the engagement of the research community and the effective integration of these advancements into future climate assessments and mitigation strategies.





**Appendix A – Opportunity processing**

The processing of opportunities proposed in the open call of August 2024, including proposals from both within the author team and more widely, was carried out by revising the evaluation made within each thematic author team in the
framework of a cross-thematic meeting in mid-September 2024. The meeting participants selected certain opportunities, rejected some, and merged others with shared scientific objectives and domain. In a subsequent step, an interactive discussion was held between members of our author team, opportunity proposal leaders, and the relevant domain communities. The goal was to harmonize the initially proposed opportunities and improve their description and data requirements. The following table summarizes the key processing actions and decisions.


| Action taken | Description | Meeting decision made | Notes from consultation and cross thematic | Notes from Author team |
|---|---|---|---|---|
| ACCEPTED | | | | |
| ID 13 | Causality of Polar Amplification | Author team meeting 06-11-2024 | Recommendation for inclusion of relevant ocean variables. | All relevant ocean variables were added. |
| ID 24 | Advancing Wind Wave Climate Modelling for Coastal Zone Dynamics, Impacts, and Risk Assessment | Author team meeting 19-09-2024 | Noted need for high-frequency surface conditions. Suggestion to merge ID 57 (Risk Assessment for offshore wind farm installation – see Impacts & Adaptation) and ID 68 (Wind driven Ocean Surface Waves). | ID 57 (see below) merged into this opportunity and the two additional variables added to the *cowclip_wind_wave _variables* group. ID 68 to remain separate (see below). |
| ID 47 | Ocean Changes, Drivers and Impacts | Author sub-group meeting 03-02-2025 | Deferred to thematic team pending further variables inclusion. | Variable groups confirmed, refined and new variables |



| | | | Suggestion to merge ID 56 (see below). | added. ID 56 relevant variables included. |
|---|---|---|---|---|
| ID 49 | Ocean Extremes | Author team meeting 06-11-2024 | Deferred to thematic team for further variable development and inclusion and merge with ID 62 (SurgeMIP storm surge intercomparison - see Appendix B of Ruane et al., 2025). | Merge with ID 62 completed and new physical parameters (and associated variables) added. |
| ID 51 | Paleoclimate Research at the Interface between Past, Present, and Future | Author team meeting 02-10-2024 | Deferred to thematic team with suggestion to merge with ID 52 (see below). | Merge with ID 52 completed. Cross thematic discussion on albedo variables. |
| ID 68 | Effects and Feedbacks of Wind-Driven Ocean Surface Waves Coupled Within Earth System Models | Author sub-group meeting 03-02-2025 | Deferred to thematic team with suggestion to merge with ID 24 (see above). | ID 24 concerns offline wave models, whereas the focus here is on coupled ESM wave components. Opportunity title updated from original Wind driven Ocean Surface Waves to reflect. |
| ID 73 | Sea Ice Changes, Drivers and Impacts | Author team meeting 06-09-2024 | Query on whether all baseline variable groups required. | *baseline_monthly* is included as it includes basic atmosphere (sat, slp, |





| | | | | u, v, etc.) and ocean (sst, u, v, mixed layer depth, etc.) variables needed to understand the drivers of sea ice changes. |
|---|---|---|---|---|
| | | MERGED | | |
| ID 46 | Ocean Assessment Reports | Author team meeting 02-10-2024 | Deferred to thematic team, more justification required. Relevance to ID 55 (Rapid Evaluation Framework - see Dingley et al., 2025). | Author team decision to merge with ID 47 (see above). |
| ID 50 | Ocean Model Intercomparison | Author team meeting 02-10-2024 | Deferred to thematic team for review after further variables submission. | Author team decision to merge with ID 47 (see above). |
| ID 52 | Paleodata Assimilation | Author team meeting 02-10-2024 | Suggestion to merge with ID 51 (see above). | Merge completed with ID 51. |
| ID 56 | Researching Stability of Meridional Overturning Circulation under the Impact of Various Forcings and Climate Trajectories | Author team meeting 19-09-2024 | Suggestion to merge with ID 47 (see above). | Author team decision to merge with ID 47 (see above). |



**Table A1:** Key processing actions and decisions, outcomes, and the dates actions were taken.

**Appendix B – New variable description**

The variables that are newly introduced in CMIP7 are tabulated below. The Coordinate Specifications column lists special aspects of the time and spatial requirements for each variable. The full grid specifications can be found in v1.2 of the CMIP7 Data Request (Data Request Task Team, 2025b).

| Physical parameter name | CF standard name | Title | Description + further detail to aid compute | Coordinate specifications |
|---|---|---|---|---|
| **absscint** | integral_wrt_depth_of_sea_water_absolute_salinity_expressed_as_salt_mass_content | Integral with respect to depth of sea water absolute salinity expressed as salt mass content | This integrated quantity is designed to be compared to observational products. It is closely aligned with the calculations of ocean heat budget below. It is a vertical integral of the absolute salinity, between layers. | longitude, latitude, oplayer4, time |
| **bigthetao200** | sea_water_conservative_temperature | Sea Water Conservative Temperature at 200 meters | Sea water conservative temperature at 200 meters. This quantity is to be provided in models using the TEOS-10 equation of state. | longitude, latitude, time, op20bar |
| **chcint** | integral_wrt_depth_of_sea_water_conservative_temperature_expressed_as_heat_content | Vertically Integrated Seawater Conservative Temperature Expressed as Heat Content | This integrated quantity is designed to be compared to observational products. It is a vertical integral of the conservative temperature (bigthetao), between layers. | longitude, latitude, oplayer4, time |
| **chl200** | mass_concentration_of_phytoplankton_expressed_as_chlorophyll_in_sea_water | Mass Concentration of Total Phytoplankton Expressed as Chlorophyll in Sea Water | Sum of chlorophyll from all phytoplankton group concentrations at 200 meters. In most models, this is equal to chldiat+chlmisc, that is the sum of Diatom Chlorophyll Mass Concentration and Other Phytoplankton Chlorophyll Mass Concentration. | longitude, latitude, time, op20bar |



| | | at 200 meters | | |
|---|---|---|---|---|
| **depthl** | depth | Depth of Lake Below the Surface | Depth of lakes, if this quantity is present in the model. If computed via volume and area, then this is lake volume divided by lake area. | longitude, latitude |
| **depthsl** | depth | Total (Cumulative) Thickness of All Soil Layers | Total (cumulative) thickness of all soil layers. This is the sum of individual thicknesses of all soil layers. | longitude, latitude |
| **dxto** | cell_x_length | Cell Length in the X Direction at t-points | The linear extent of the cell in the x direction of the horizontal grid centered at t-points (points for tracers such as temperature, salinity, etc.). Not applicable to unstructured grids. | longitude, latitude |
| **dxuo** | cell_x_length | Cell Length in the X Direction at u-points | The linear extent of the cell in the x direction of the horizontal grid centered at u-points (points for velocity in the x-direction). Not applicable to unstructured grids. | longitude, latitude |
| **dxvo** | cell_x_length | Cell Length in the X Direction at v-points | The linear extent of the cell in the x direction of the horizontal grid centered at v-points (points for velocity in the y-direction). Not applicable to unstructured grids. | longitude, latitude |
| **dyto** | cell_y_length | Cell Length in the Y Direction at t-points | The linear extent of the cell in the y direction of the horizontal grid centered at t-points (points for tracers such as temperature, salinity, etc.). Not applicable to unstructured grids. | longitude, latitude |
| **dyuo** | cell_y_length | Cell Length in the Y Direction at u-points | The linear extent of the cell in the y direction of the horizontal grid centered at u-points (points for velocity in the x-direction). Not applicable to unstructured grids. | longitude, latitude |
| **dyvo** | cell_y_length | Cell Length in the Y Direction at v-points | The linear extent of the cell in the y direction of the horizontal grid centered at v-points (points for velocity in the y-direction). Not applicable to unstructured grids. | longitude, latitude |
| **hfacros sline** | ocean_heat_tr ansport_across _line | Ocean Heat Transport across Lines | Depth-integrated total heat transport from resolved and parameterized processes across different lines on the Earth's surface (based on appendix J and table J1 of Griffies et al., 2016). Formally, this means the integral along the line of the normal component of the heat transport. Positive and negative numbers refer to total northward/eastward and southward/westward | oline, time |

 

| | | | transports, respectively. The transport should be evaluated for the full depth of the ocean, except for the Pacific Equatorial Undercurrent, which is averaged from the surface to 350m. Use Celsius for temperature scale. | |
|---|---|---|---|---|
| **hfxint** | ocean_heat_x _transport | Vertically Integrated Ocean Heat X Transport | Ocean heat x transport vertically integrated over the whole ocean depth. Contains all contributions to 'x-ward' heat transport from resolved and parameterized processes. Use Celsius for temperature scale. Report on native horizontal grid. Note that this variable was called hfx in CMIP6; hfx in CMIP7 now represents the 3D ocean heat x transport. | longitude, latitude, time |
| **hfyint** | ocean_heat_y _transport | Vertically Integrated Ocean Heat Y Transport | Ocean heat y transport vertically integrated over the whole ocean depth. Contains all contributions to 'y-ward' heat transport from resolved and parameterized processes. Use Celsius for temperature scale. Report on native horizontal grid. Note that this variable was called hfy in CMIP6; hfx in CMIP7 now represents the 3D ocean heat x transport. | longitude, latitude, time |
| **mpw** | sea_surface_w ave_mean_per iod | Total Wave Mean Period | Average wave period (i.e., time in-between two wave crests) across the entire two-dimensional wave spectrum, incorporating both wind-sea and swell waves. In spectral wind wave models, it is calculated using spectral moments, mathematical measures that describe the shape and characteristics of the wave spectrum. | longitude, latitude, time |
| **mpwwi ndsea** | sea_surface_w ind_wave_me an_period | Wind Sea Wave Mean Period | Average wave period (i.e., time in-between two wave crests) of wind-sea waves only (i.e., local wind waves). In spectral wind wave models, it is calculated using spectral moments, mathematical measures that describe the shape and characteristics of the wave spectrum. | longitude, latitude, time |
| **mpwsw ell** | sea_surface_s well_wave_m ean_period | Swell Wave Mean Period | Average wave period (i.e., time in-between two wave crests) of swell waves only (i.e., waves that have propagated away from their generation area). In spectral wind wave models, it is calculated using spectral moments, mathematical measures that describe the shape and characteristics of the wave spectrum. | longitude, latitude, time |



| o2200 | mole_concentration_of_dissolved_molecular_oxygen_in_sea_water | Dissolved Oxygen Concentration at 200 meters | Dissolved oxygen concentration at 200 meters. This quantity is to be calculated in models with a biogeochemistry package calculating an oxygen budget. | longitude, latitude, time, op20bar |
|---|---|---|---|---|
| pfscint | integral_wrt_depth_of_sea_water_preformed_salinity_expressed_as_salt_mass_content | Vertically Integrated Seawater Preformed Salinity Expressed as Salt Mass Content | This integrated quantity is designed to be compared to observational products. It is a vertical integral of the preformed salinity, between layers. | longitude, latitude, oplayer4,time |
| phcint | integral_wrt_depth_of_sea_water_potential_temperature_expressed_as_heat_content | Integrated Ocean Heat Content from Potential Temperature | This integrated quantity is designed to be compared to observational products. It is a vertical integral of the potential temperature (thetao), between layers, expressed in energy units. | longitude, latitude, oplayer4,time |
| rsdsis | surface_downwelling_shortwave_flux_in_air | Surface Downwelling Shortwave Radiation over Ice Sheets | Surface Downwelling Shortwave Radiation over the ice-sheet covered portion of a grid cell, including snow. Can be used for computation of surface albedo. | longitude, latitude, time |
| rsdslni | surface_downwelling_shortwave_flux_in_air | Surface Downwelling Shortwave Radiation over Land Not Covered by Ice Sheets or Snow | Surface Downwelling Shortwave Radiation over the portion of a land grid cell not covered by ice sheets or snow. Can be used for computation of surface albedo. | longitude, latitude, time |
| rsdsoni | surface_downwelling_shortwave_flux_in_air | Surface Downwelling Shortwave Radiation over Ocean Not Covered by Sea Ice | Surface Downwelling Shortwave Radiation over the portion of an ocean grid cell not covered by sea ice. Can be used for computation of surface albedo. | longitude, latitude, time |
| rsdss | surface_down | Surface | Surface Downwelling Shortwave Radiation over | longitude, latitude, |



| | welling_short wave_flux_in _air | Downwelling Shortwave Radiation over Snow | the portion of a land grid cell covered by snow but not by ice. Can be used for computation of surface albedo. | time |
|---|---|---|---|---|
| **rsdssi** | surface_down welling_short wave_flux_in _air | Surface Downwelling Shortwave Radiation over Sea Ice | Surface Downwelling Shortwave Radiation over the portion of an ocean grid cell covered by sea ice, including snow. Can be used for computation of surface albedo. | longitude, latitude, time |
| **rsusis** | surface_upwel ling_shortwav e_flux_in_air | Surface Upwelling Shortwave Radiation over Ice Sheets | Surface Upwelling Shortwave Radiation over the ice-sheet covered portion of a grid cell, including snow. Can be used for computation of surface albedo. | longitude, latitude, time |
| **rsuslni** | surface_upwel ling_shortwav e_flux_in_air | Surface Upwelling Shortwave Radiation over Land Not Covered by Ice Sheets or Snow | Surface Upwelling Shortwave Radiation over the portion of a land grid cell not covered by ice sheets or snow. Can be used for computation of surface albedo. | longitude, latitude, time |
| **rsusoni** | surface_upwel ling_shortwav e_flux_in_air | Surface Upwelling Shortwave Radiation over Ocean Not Covered by Sea Ice | Surface Upwelling Shortwave Radiation over the portion of an ocean grid cell not covered by sea ice. Can be used for computation of surface albedo. | longitude, latitude, time |
| **rsuss** | surface_upwel ling_shortwav e_flux_in_air | Surface Upwelling Shortwave Radiation over Snow | Surface Upwelling Shortwave Radiation over the portion of a land grid cell covered by snow. Can be used for computation of surface albedo. | longitude, latitude, time |
| **rsussi** | surface_upwel ling_shortwav e_flux_in_air | Surface Upwelling Shortwave Radiation over Sea Ice | Surface Upwelling Shortwave Radiation over the portion of an ocean grid cell covered by sea ice, including snow. Can be used for computation of surface albedo. | longitude, latitude, time |
| **scint** | integral_wrt_d | Vertically | This integrated quantity is designed to be | longitude, latitude, |



| | epth_of_sea_water_practical_salinity_expressed_as_salt_mass_content | Integrated Seawater Practical Salinity Expressed as Salt Mass Content | compared to observational products. It is a vertical integral of the practical salinity, between layers. | oplayer4,time |
|---|---|---|---|---|
| **sduo** | sea_surface_wave_stokes_drift_eastward_velocity | Eastward Surface Stokes Drift | The eastward component of the net drift velocity of ocean water caused by surface wind-sea waves. The Stokes drift velocity could be defined as the difference between the average Lagrangian flow velocity of a fluid parcel, and the average Eulerian flow velocity of the fluid at a fixed position. | longitude, latitude, time |
| **sdvo** | sea_surface_wave_stokes_drift_northward_velocity | Northward Surface Stokes Drift | The northward component of the net drift velocity of ocean water caused by surface wind-sea waves. The Stokes drift velocity could be defined as the difference between the average Lagrangian flow velocity of a fluid parcel, and the average Eulerian flow velocity of the fluid at a fixed position. | longitude, latitude, time |
| **sfacrossline** | ocean_salt_mass_transport_across_line | Ocean Salt Mass Transport across Lines | Depth-integrated total salt mass transport from resolved and parameterized processes across different lines on the Earth's surface (based on appendix J and table J1 of Griffies et al., 2016). Formally, this means the integral along the line of the normal component of the heat transport. Positive and negative numbers refer to total northward/eastward and southward/westward transports, respectively. The transport should be evaluated for the full depth of the ocean, except for the Pacific Equatorial Undercurrent, which is averaged from the surface to 350m. | oline, time |
| **sftlkf** | area_fraction | Fraction of the Grid Cell Occupied by Lake | Fraction of horizontal land grid cell area occupied by lake. | longitude, latitude, typelkins |
| **sfx** | ocean_salt_x_transport | 3D Ocean Salt Mass X Transport | Contains all contributions to 'x-ward' salt mass transport from resolved and parameterized processes. Report on native horizontal grid. | longitude, latitude, olevel, time |
| **sfxint** | ocean_salt_x_ | Vertically | Ocean salt mass x transport vertically integrated | longitude, latitude, |





| | transport | Integrated Ocean Salt Mass X Transport | over the whole ocean depth. Contains all contributions to 'x-ward' salt mass transport from resolved and parameterized processes. Report on native horizontal grid. | time |
|---|---|---|---|---|
| **sfy** | ocean_salt_y_ transport | 3D Ocean Salt Mass Y Transport | Contains all contributions to 'y-ward' salt mass transport from resolved and parameterized processes. Report on native horizontal grid. | longitude, latitude, olevel, time |
| **sfyint** | ocean_salt_y_ transport | Vertically Integrated Ocean Salt Mass Y Transport | Ocean salt mass y transport vertically integrated over the whole ocean depth. Contains all contributions to 'y-ward' salt mass transport from resolved and parameterized processes. Report on native horizontal grid. | longitude, latitude, time |
| **simpeff conc** | area_fraction | Fraction of Sea Ice Covered by Effective Melt Pond | Area fraction of sea-ice surface that is covered by open melt ponds, that is melt ponds that are not covered by snow or ice lids. This represents the effective (i.e. radiatively-active) melt pond area fraction. | longitude, latitude, time, typemp |
| **sisnmas sn** | surface_snow _mass | Sea-Ice Snow Mass North | Total integrated mass of snow on sea ice in the Northern Hemisphere grid cells | time |
| **sisnmas ss** | surface_snow _mass | Snow Mass on Sea Ice South | Total integrated mass of snow on sea ice in the Southern Hemisphere grid cells | time |
| **swh** | sea_surface_w ave_significan t_height | Total Significant Wave Height | Average height of the highest one-third of waves present in the sea state, incorporating both wind-sea and swell waves. This is a key parameter for describing wave energy and is derived from the wave spectrum using spectral moments. Specifically, this parameter is four times the square root of the integral over all directions and all frequencies of the two-dimensional wave spectrum. | longitude, latitude, time |
| **swhmax** | sea_surface_w ave_significan t_height | Maximum Significant Wave Height | Highest value of the significant wave height simulated within a given time range (e.g., daily or monthly). The significant wave height (swh) is derived from the wave spectrum using spectral moments. Specifically, swh is four times the square root of the integral over all directions and all frequencies of the two-dimensional wave spectrum. | longitude, latitude, time |
| **swhwin dsea** | sea_surface_w ind_wave_sig | Wind Sea Significant | Average height of the highest one-third of waves present in the sea state, incorporating just wind- | longitude, latitude, time |

   



| | nificant_heigh t | Wave Height | sea waves (i.e., local wind waves). It is derived from the wind-sea wave spectrum using spectral moments. Specifically, this parameter is four times the square root of the integral over all directions and all frequencies of the two-dimensional wind-sea wave spectrum. | |
|---|---|---|---|---|
| **swhswel l** | sea_surface_s well_wave_si gnificant_heig ht | Swell Significant Wave Height | Average height of the highest one-third of waves present in the sea state, incorporating just swell waves (i.e., waves that have propagated away from their generation area). This parameter is derived from all swell partitions of the wave spectrum using spectral moments. Specifically, this parameter is four times the square root of the integral over all directions and all frequencies of the components of the two-dimensional wave spectrum that are not under the influence of local wind. | longitude, latitude, time |
| **thetao2 00** | sea_water_pot ential_tempera ture | Sea Water Potential Temperature at 200 meters | Sea water potential temperature at 200 meters. This quantity is to be provided in models using the older equations of state. | longitude, latitude, time, op20bar |
| **thkcellu o** | cell_thickness | Ocean Model Cell Thickness at u-points | The time varying thickness of ocean cells centered at u-points (points for velocity in the x-direction). "Thickness" means the vertical extent of a layer. "Cell" refers to a model grid-cell. | longitude, latitude, olevel, time |
| **thkcellv o** | cell_thickness | Ocean Model Cell Thickness at v-points | The time varying thickness of ocean cells centered at v-points (points for velocity in the y-direction). "Thickness" means the vertical extent of a layer. "Cell" refers to a model grid-cell. | longitude, latitude, olevel, time |
| **uos** | surface_sea_w ater_x_velocit y | Surface Sea Water X Velocity | This variable is standard in most models | longitude, latitude, time depth0m |
| **vos** | surface_sea_w ater_y_velocit y | Surface Sea Water Y Velocity | This variable is standard in most models | longitude, latitude, time depth0m |
| **wdir** | sea_surface_w ave_from_dire ction | Total Wave Direction | Mean direction of wave propagation (direction from which the wave is coming) derived from the total wave energy spectrum, incorporating both wind-sea and swell waves. This variable is usually expressed in degrees relative to true north. | longitude, latitude, time |



| **wdirwindsea** | sea_surface_wind_wave_from_direction | Wind Sea Wave Direction | Mean direction of wave propagation (direction from which the wave is coming) derived from the wind-sea component of the wave energy spectrum (i.e., local wind waves). This variable is usually expressed in degrees relative to true north. | longitude, latitude, time |
| --- | --- | --- | --- | --- |
| **wdirswell** | sea_surface_swell_wave_from_direction | Swell Wave Direction | Mean direction of wave propagation (direction from which the wave is coming) derived from the swell component of the wave energy spectrum (i.e., waves that have propagated away from their generation area). This variable is usually expressed in degrees relative to true north. | longitude, latitude, time |
| **wpdir** | sea_surface_wave_from_direction_at_variance_spectral_density_maximum | Total Peak Wave Direction | Direction of wave propagation (direction from which the wave is coming) derived from the total wave energy spectrum, incorporating both wind-sea and swell waves, by identifying the direction associated with the peak (maximum) energy density. This variable is usually expressed in degrees relative to true north. | longitude, latitude, time |
| **wpdirwindsea** | sea_surface_wind_wave_from_direction_at_variance_spectral_density_maximum | Wind Sea Peak Wave Direction | Direction of wave propagation (direction from which the wave is coming) derived from the wind-sea component of the wave energy spectrum (i.e., local wind waves), by identifying the direction associated with the peak (maximum) energy density. This variable is typically expressed in degrees relative to true north. | longitude, latitude, time |
| **wpdirswell** | sea_surface_swell_wave_from_direction_at_variance_spectral_density_maximum | Swell Peak Wave Direction | Direction of wave propagation (direction from which the wave is coming) derived from the swell component of the wave energy spectrum (i.e., waves that have propagated away from their generation area), by identifying the direction associated with the peak (maximum) energy density. This variable is typically expressed in degrees relative to true north. | longitude, latitude, time |
| **wpp** | sea_surface_wave_period_at_variance_spectral_density_maximum | Total Wave Peak Period | Wave period associated with the most energetic waves in total wave spectrum, incorporating both wind-sea and swell waves. In spectral wind wave models, this represents the spectral peak across the entire two-dimensional wave | longitude, latitude, time |





| | | | spectrum, incorporating both wind-sea and swell waves. | |
| --- | --- | --- | --- | --- |
| **wppwindsea** | sea_surface_wind_wave_period_at_variance_spectral_density_maximum | Wind Sea Wave Peak Period | Wave period associated with the most energetic wind-sea waves (i.e., local wind waves). In spectral wind wave models, this represents the spectral peak across part of the two-dimensional wave spectrum, incorporating just wind-sea waves. | longitude, latitude, time |
| **wppswell** | sea_surface_swell_wave_period_at_variance_spectral_density_maximum | Swell Wave Peak Period | Wave period associated with the most energetic swell waves (i.e., waves that have propagated away from their generation area). In spectral wind wave models, this represents the spectral peak across part of the two-dimensional wave spectrum, incorporating just swell waves. | longitude, latitude, time |

**Table B1:** New variables introduced to CMIP in this data request.



**Code and data availability**

The variables and their metadata included latest CMIP7 Assessment Fast Track Data Request can be accessed
at https://doi.org/10.5281/zenodo.15288187 (Data Request Task Team, 2025b). At the time of this publication, the
latest major release is v1.2 (Data Request Task Team, 2025a; https://doi.org/10.5281/zenodo.15116894), and the latest
minor release is v1.2.1 (Data Request Task Team, 2025b; https://doi.org/10.5281/zenodo.15288187).

**Author contributions**

BFK and PDR led the conceptualization, investigation, methodology, and data curation with contributions from AMT,
CS, EOR, CM, AM, YA, PJD, NF, VH, ALM, FM, JM, DS, PCT, WLT and MV. BFK and PDR led the writing of this
manuscript with support in writing of the original draft from AMT, CS, EOR, AM, NF, VH, CH, DI, GM, ALM, FM,
DS, PCT and WLT and review and editing support from YA, PJD, and MV. MK provided coordination support to the
Earth System theme and EOR provided resources and project administration support.

**Competing interests**

The authors declare that they have no conflict of interest.

**Acknowledgments**

The Oceans and Sea Ice Theme Author Team acknowledges the valuable contributions from a widespread scientific
community who participated in the data request effort and public consulting processes. We thank Martin Juckes, Robert
Fajber, and Tommi Bergman for helpful comments on the draft. BFK is funded by NSF RISE-2425380, the Equitable
Climate Futures initiative at Brown University, and the SASIP project of the Schmidt Futures Foundation. PDR is
funded by the European Union (ERC, ArcticWATCH, 101040858). AMT is funded by the European Union (EERIE
project, Grant Agreement No 101081383). CS is funded through the Alfred Wegener Institute's research program
"Changing Earth – Sustaining our Future" of the Helmholtz Association, the Helmholtz Climate Initiative REKLIM and
acknowledges support by the European Union (ERC, i2B, 101118519). The CMIP IPO (EOR) is hosted by the
European Space Agency, with staff provided on contract by HE Space Operations Ltd. AM contributed voluntarily with
the support of CSIRO and the University of Melbourne. YA acknowledges funding support from the European Union's
project EPOC "Explaining and Predicting the Ocean Conveyor", EU grant 101059547 and UKRI grant 10038003, and
from the EC Horizon Europe project OptimESM "Optimal High Resolution Earth System Models for Exploring Future



Climate Changes" under grant 101081193 and UKRI grant 10039429, and from the UK NERC Highlight Topic
"Interacting ice Sheet and Ocean Tipping - Indicators, Processes, Impacts and Challenges (ISOTIPIC)", grant
NE/Y503320/1 and funding support from the UK NERC LTS-M projects BIOPOLE (NE/W004933/1) and CANARI
(NE/W004984/1), and from Atlantic Climate and Environment Strategic Science (AtlantiS) grant (NE/Y005589/1). The
work of PJD from Lawrence Livermore National Laboratory (LLNL) is supported by the Regional and Global Model
Analysis (RGMA) program area under the Earth and Environmental System Modeling (EESM) program within the
Earth and Environmental Systems Sciences Division (EESSD) of the United States Department of Energy's (DoE)
Office of Science. This work was performed under the auspices of the US DoE by LLNL under contract DE-AC52-
07NA27344. LLNL IM Release: LLNL-JRNL-2007705. NF is funded under U.S. Department of Energy (DOE) Award
Number DE-SC0023070. CH is funded under Swedish National Space Agency grant number 2022-00149. DI is
supported by the Foundation Euro-Mediterranean Centre on Climate Change (CMCC). GM is supported by NERC
through the ALPACA (NE/Y005279/1) project and by UKRI (10039018) as part of the EPOC project (Explaining and
Predicting the Ocean Conveyor; Grant 101059547). ALM is partially supported by the project "Development of the
Community Earth System Model - MONAN" under the contract No. 01340.005344/2021-50. FM is a F.R.S.-FNRS
Research Associate. JM acknowledges funding support from the EC Horizon Europe project OptimESM "Optimal High
Resolution Earth System Models for Exploring Future Climate Changes" under grant 101081193 and UKRI grant
10039429, and from the UK NERC Highlight Topic "Interacting ice Sheet and Ocean Tipping - Indicators, Processes,
Impacts and Challenges (ISOTIPIC)", grant NE/Y503320/1. PCT is supported by the NASA Radiation Budget Science
Project and NASA's Radiation Sciences and Cryosphere Programs. WLT is supported by the National Science and
Technology Council, Taiwan (NSTC 113-2111-M-002-016 -). MK is supported by the MEXT SENTAN Program
(Grant Number JPMXD0722681344). For all the EU projects, views and opinions expressed are however those of the
author(s) only and do not necessarily reflect those of the European Union or the European Research Council Executive
Agency, European Climate Infrastructure and Environment Executive Agency (CINEA), or European Research Council
Executive Agency. Neither the European Union nor the granting authority can be held responsible for them.



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
