# Peer review of "CMIP7 Data Request: Ocean and Sea Ice Priorities and Opportunities"

_EGUsphere, 2025_

## Author Comment (AC1)

Key: Blue Text=Reviewer. Black Text=Author Response. Purple Text=Alterations to the manuscript.

**Anonymous Reviewer #1**

In this paper, Baylor Fox-Kemper and his co-authors present an overview of the CMIP7 variable request for the ocean and sea ice.

Before outlining some comments on this paper, I first want to express my gratitude to the authors for taking on this huge effort which I see this as a truely outstanding service to our community. Thank you!

Thank you for the kind words. It is our honor to have been selected to serve the community in this role, and we aspire to provide a clear and compelling set of documents.

As regards to the paper itself, I find this a bit difficult to review as it's not a classical scientific paper, but instead more of an outline of the result of a community effort to define overarching questions that could/should be addressed with CMIP7 model output. As such, I'm a bit unsure as to which level of criticism/comment is warranted: The paper is scientifically formally correct and can thus in my view be published as is (with some very minor edits as outlined below).

We understand and appreciate the reviewer's effort in any case to improve the paper.

However, as a reader, I also felt that a few important questions remained open, but I would like to leave it to the authors to decide to which degree these can be answered within the scope of this publication. I therefore just list these here and look forward to seeing maybe at least some level of discussion in a revised version of this paper.

Thank you. Our detailed point-by-point responses are given below.

1. I would have wished for more "lessons learned" from CMIP6, and for some discussion as to how these lessons were addressed in CMIP7. In particular, I feel that the data request for CMIP6 was already overwhelming, and now even more variables are added to that list. What did we learn more concretely about the variable usage and efficiency of CMIP6 output, and how was this considered here? Which variables or groups of variables are dropped for CMIP7, or are we simply requesting more from the modelling centers?

The variable selection process was guided by the CMIP IPO, which brought forward principles, scheduled meetings, and so on. Our team divided the different tasks, consulted our communities, considered the variables we used in previous documents and assessments, and brought those discussions together to try and synthesize a whole.

The principles here were to reduce the data volume (where possible), the number of variables (if possible), and to clarify the rationales for the inclusion of each variable. We note that this document is only a data request rather than a demand, so modeling centers must be persuaded to provide the variables through our text, their own experience, etc.

In the course of this work, we worked with members of the CMIP WIP (WCRP Infrastructure Panel) who had worked extensively on the previous data request and with the access statistics to the CMIP6 archive. Unfortunately, there was a key flaw in the collection of this data, which is that when an institution downloaded a dataset once, that was recorded, but if it was shared widely from that download (as is common practice at many institutions and modeling centers), only the initial download appears in the statistics. We think this severely undercounts the usage of some of the main variables. However, we were able to examine which variables were not uploaded often, which indicates that they were not well-justified in prioritization, and we also were able to downgrade their prioritization level or remove them entirely.

Of course, lessons learned from CMIP6 may be subjective (although they informed our discussions and writing to a fair degree). From the side of modelling centers, as the reviewer remarks, the CMIP6 data request, that included more than 1200 MIP variables and 2000 distinct CMOR variables (Juckes et al., 2020), may have been seen as very demanding for modelling centers when aligning preparation and publication of model variables with resource limitations in simulation production and CMORization. On the other hand, we see that the variety of research interests in CMIP is becoming larger rather than smaller with time (climate extremes, ocean waves, coupled ice sheets, variable land surface characteristics, etc.), necessitating adding to, or refocusing, elements of the data request. It is in these applications that most of our effort in the writing process was expended.

One lesson that has guided the preparation of the CMIP7 data request is that previous data requests could have benefited from more structure and from a clearer illustration of links between requested variables and specific research fields and scientific questions, that would motivate the production of specific model variables. Juckes et al. (2025) prepare the basic foundation for data production at modelling centers for Earth System Research and model-intercomparison with a baseline data request that is very concise (135 variables only) but that arguably would limit the versatility of CMIP7 output for the wider scientific community. Therefore, guided by CMIP6 variable download statistics and by input from the various Earth System research communities, the various thematic groups extensively reflected on the need for additional model output. As a general rule, a model variable has only been considered in the CMIP7 data request if its production has been scientifically motivated by at least one of the various opportunities that describe data demands for specific research fields or scientific questions (this framework was provided to us by the CMIP leadership). While this justifies a variable to be produced, prioritization may of course still lead to different evaluation of variable relevance at modelling centers.

In the end, the combined effort of baseline request and author theme opportunities culminated а collection 1991 CMIP7 in of variables in the data request (https://airtable.com/appOcSa4gXyzHThmm/shrkayKObes58Zu45/tblpo5L8maBIGIM1B/viwNNz rgK5oPL7zk2; accessed 8th of August, 2025). While this still involves a large volume of model output, we hope that formulation of opportunities, organization of variables into thematic groups, and categorization of priority will help modelling centers to guide output production, and, where necessary, prioritization, following their own capabilities and scientific interests.

We also note, specifically in the "outstanding gaps" section 4.2 and "key reflections" section 4.3, that the process we were given was largely manual (as opposed to automated) and based on community discussions. The latter is extremely valuable, of course, but is unlikely to reduce data volumes. The former is truly a missed opportunity. We hope that during the CMIP7 usage cycle that new tools, such as emulators and better post-processing diagnostic packages, and even AI data mining of the literature based on CMIP7, will enable automated tools can make future data requests more meaningful and more easily accomplished. We hope that this section of the paper, as well as other perspective papers written by our team members and others on these topics, will improve our community efforts going forward.

2. What are the pros and cons of the approach of defining "opportunities"? Do the authors feel it is warranted that 2 out of 7 of these are related to waves? Given that the two overarching opportunities of the ocean and the sea ice variables seem to mirror the approach taken in CMIP6, the introduction of opportunities seems to make the call wider rather than more concrete, which seems to go against the intention of opportunities. Some reflection on this would be helpful. Is this approach too broad/too limiting/helpful??

This is an excellent question. The "opportunities" framing was meant to be cross-cutting over multiple MIPs, whereas the CMIP6 process was organized more through the contributions of MIPs (e.g., the ocean data request papers (Griffies et al., 2016; Orr et al., 2017) were created by the OMIP and ocean biogeochemistry MIP teams, the sea ice data request paper was created by the SIMIP team, etc.). Our team thought the opportunities approach was successful for those of us who had worked on MIPs and the preceding data request process, and those of us new to the process were given a guiding principle that helped organize their contributions. Overall, it was an organizing principle that allowed us to cross-check over multiple potential users' needs.

As to the 2 wave opportunities, this was particularly interesting to our team and not of our own design. A few of our authors were involved in wave modeling efforts for CMIP6 (including Fox-Kemper), but those opportunities were contributed from outside of our team. The fact that there are two opportunities reflects the two ways that waves are simulated in CMIP, offline via the COWCLIP community (e.g., Hemer et al., 2016) and its descendants, and online via the efforts at FIO and NCAR during CMIP6 and additional teams joining in CMIP7. Because these two applications differ substantially in which data is requested (forcing vs. forcing and response, essentially), and the fact that only some modeling centers have online wave capabilities, while all modeling centers can choose to provide forcing data for offline wave calculations, we found it easier to organize them separately.

The big pro we see in the opportunity framework is bringing structure into an otherwise monolithic data request. We also think that this outweighs cons, with the only one that comes to mind being the scattering of scientific motivation for variable production across different opportunities and thematic papers. While each and every opportunity has merit for specific scientific communities, scientific preference will, of course, allow very subjective evaluation of the relevance of specific variables. We would like to note, though, that opportunities that appear in the manuscript have been reviewed and supported by multiple thematic author themes (e.g.,

ocean and sea ice, atmosphere, etc). As a result of the review and evaluation process, and in an attempt to simplify the structure and to maintain a threshold of scientific relevance, various opportunities were either merged or rejected (see Annex 1 for the ones related to ocean and sea ice). Others, which may be justified, were not proposed (tipping points and ice sheets) and are discussed elsewhere in this response. Therefore, for all variables mentioned in the various opportunities, there is justification for them to be considered in the data production process. The selection of variables for data production and research is then left to the scientific interests of involved parties, who may consider the described opportunities as guidance.

3. What is special about "paleo" variables, why are they different from the standard variables, or is the idea here to define a subset given the length of the simulations?

The paleo community has different traditions, as until CMIP7, the Paleoclimate Modelling Intercomparison Project (PMIP) was a separate activity with meetings organized outside of the CMIP process (although sharing and some PMIP scientists participating in the CMIP data requests). Many of these variables are also included in the baseline list, but some are not, even though they are critical to paleo applications (e.g., full depth datasets and special attention to overturning analysis). The PMIP-sponsored Opportunity 51, Paleoclimate research across time scales, collects all variables that support scientific research in the paleoclimate research community and in linked research fields (e.g., current and future warmer climate). In paleoclimate research, the focus is sometimes different from that of more recent and near-future climate research in CMIP. This means that variable selection addresses additional Earth System components, like permafrost, stable water isotopes, or metrics that diagnose the state of the Earth System under climates that may significantly differ from today. While we have referenced in this opportunity some research that illustrates how climate model output may improve the link between climate research of the past, the present, and the future, we acknowledge that the aim for brevity does not allow us to provide extensive justification of every variable. Nevertheless, the thematic organization of model output into the various variable groups will highlight justification of variables for specific scientific questions that are linked to, e.g., data assimilation, stable water isotopes, or stability and state of the cryosphere under different background climates.

4. Some discussion of on-line analysis versus the storage of huge amounts of data would be helpful, I find. To which degree do the authors think that we should focus more on high-level analyses that are calculated on the fly while the simulations are running, versus storing petabytes of output variables that researchers than use to calculate the same integrated metrics again and again? (I think this indeed is an open question)

This is an excellent and interesting suggestion. We have added the following text to the "Key reflections" section:

One challenge that confronts teams such as ours trying to optimize the data storage effort is that some variables can, of course, only be calculated online during simulation, while others can be calculated offline when given the correct input data. Indeed, standard analyses that may be centrally prepared, as considered for example in the CMIP7 Rapid

Evaluation Framework (Hoffman et al., 2025; Hassler et al. 2025), are ideal to limit data storage load and to speed up exploration by avoiding preparing the same analyses over and over again. Two oceanic examples that our team considered were mixed layer depth, which can only be calculated at full accuracy online (see Treguier et al., 2023, for a quantification of the errors when calculating after the fact), and ocean heat content, which can be calculated after the fact from 3D temperature, salinity, and grid specifications, but only in an approximate way when the vertical coordinate varies in time. Treguier et al. find that mixed layer depth errors in calculating after the fact were somewhat smaller than the errors in CMIP6 due to discrepancies in the definition of mixed layer depth used by multiple modeling centers. In this data request, we seek to eliminate these inter-model definitional discrepancies, which then leaves after-the-fact calculation as a leading error, which justifies our request for the mixed layer depth as calculated online. The ocean heat content (and salt content) can be calculated accurately from 3D fields for some vertical coordinates and only approximately for others, but this calculation is done so often and is so expensive in data recall that we ask modeling centers to do it in advance. Providing the calculated ocean heat content necessarily increases the number of variables requested from modeling centers, with some redundancy in many cases, but as these variables are used widely, require downloading large data volumes, and are prone to error in calculation, we deem this addition to be justified. Another example is the request for modeling centers to provide hemispherically-integrated sea ice variables (such as sea-ice area, extent, volume, etc.), as we know from previous CMIP phases that these are the most downloaded sea ice variables, and calculating them offline after regridding can lead to errors compared to doing it online. In general, there tends to be a tradeoff between the number of variables requested (which can actually reduce the data volume needed to be downloaded and stored when they are, e.g., integrated as ocean heat content is or 2D as mixed layer depth is) and the volume of data requested (e.g., higher frequency 3D fields reduce the rectification errors in after-the-fact calculations in variable vertical grid models). We only anticipate these issues will become more complex as model configurations diversify in vertical grid specifications and unstructured horizontal grids. The OMIP protocol and diagnostics under discussion, which will be submitted for review soon (Fox-Kemper, personal communication), will contain further details and discussion.

A. Treguier, C. de Boyer Montegut, A. Bozec, E. P. Chassignet, B. Fox-Kemper, A. M. Hogg, D. Iovino, A. E. Kiss, J. L. Sommer, Y. Li, P. Lin, C. Lique, H. Liu, G. Serazin, D. Sidorenko, Q. Wang, X. Xu, and S. Yeager. The mixed layer depth in the ocean model intercomparison project (OMIP): Impact of resolving mesoscale eddies. *Geoscientific Model Development*, 16(13):3849--3872, July 2023.

5. Some reflection on the time line for CMIPs would have been nice to have included here from a specific sea-ice and ocean perspective. From the discussion, I understand that we might be overwhelming ourselves with the current pace of CMIP activities - would the authors have concrete suggestions for improving this situation for our communities?

The baseline variable paper contains some of this information, but as is normal, the CMIP schedule involves unavoidable delays that are hard to anticipate. The CMIP7 simulations that are not part of the Assessment Fast Track do not have a fixed schedule at this time (e.g., OMIP)

and most of PMIP). A historical perspective on the timelines proposed and achieved, such as Durack et al. (2025), is helpful context. We believe that regular model-intercomparison exercises enable early identification of promising and less promising routes in ongoing model development. Furthermore, via ESGF, they provide an ensemble of model data that is of great value to the research community. Therefore, CMIP and related activities have their justification.

6. Which concrete criteria were used to define the priorities for variable requests? How are they different from those in CMIP6?

These priorities were decided through discussions among our team, together with community input. In most cases, they are the same as in CMIP6, except where usage and delivery statistics indicated that they were actually interpreted by modeling centers and users to be lower priority than the CMIP6 data request prioritization indicated.

7. Are there variables that are requested to be supplied as a group given that for example supplying just 60 % of budget variables often means that even those 60 % cannot be used if the other 40 % of variables are not supplied

Not strictly. We do note that this data request is primarily targeted for the Assessment Fast Track, which we do not anticipate will define the extent of variables studied during model development at most modeling centers, but will be only a subset of those variables. Thus, closure of budgets is not a goal of this data request in most cases (or is low priority), although some of the refinements suggested in sea ice albedo variables were intended to improve the potential for budgets to be closed accurately. Nonetheless, modeling centers where closing budgets is a priority will find that they are able to provide those variables if they choose to, although we do not guarantee that this will be as accurate as budgets that are closed in online calculations, where, e.g., rectification effects from vertical grid variations can be accounted for entirely.

However, certain budgets can be calculated, and these are provided in the same opportunity and variable group (as in the sea ice opportunity which has a focus on albedo budgets). The opportunities are intended to argue for groupings of variables that would be convenient to have together, with commensurate frequency and resolution requirements. These are not all of the things one might consider doing with the models, so of course, there are gaps, some of which are specified in section 4.2.

8. While the term "tipping" is mentioned in the abstract, there is no dedicated opportunity related to the stability of e.g. the AMOC or other large-scale ocean features. Is there a reason for why this is not targeted explicitly?

No team proposed an opportunity along these lines, although we certainly might have expected, e.g., the TIPMIP team to have done so. However, our "Ocean Changes, Drivers and Impacts" opportunity includes a specific variable group for the meridional overturning streamfunction. Furthermore, our paleoclimate variables include the standard variables used for tipping point assessment, as that is a common topic of study in those applications. Specific questions, like

TIPMIP model output, are planned to fall under the future unharmonized data request with the descriptor "updates as needed by MIPs".

9. The inclusion of physical vs. non-physical variables is unclear. I first thought that this paper only dealt with physical variables but then saw that for example chl200 is requested which relates to chlorophyll. Some discussion of physics vs. biogeochemistry vs. biology of this variable request would be helpful

We focus on physical variables, but in that specific case, the literature is examining multi-factor compound extremes. The biogeochemistry effort will also request chlorophyll, but at other depths and different temporal frequencies. In the case of compound extremes, the frequency must match across the multiple factors. Similarly, stable isotopes are part of the paleoclimate opportunity, but these are not physical variables but have direct relevance to the physical questions at hand, especially the model-observation/proxy comparisons.

10. I was surprised to find that lakes are included in this request. Was this also the case in CMIP6? Some discussion on lakes vs. seas vs. oceans would be helpful, also related to the modeling of the hydrological cycle over land grid cells.

The inclusion of lakes is a model-dependent choice. We only request lake depth for the models where this is considered part of the ocean and sea ice component. In many models, lakes are part of the hydrological modeling component. Which large lakes go into which model is not in agreement across the community. Also, depending on the land surface characteristics of specific paleoclimatic time periods, climate research across time scales will lead to the use of various (very different) land surface characteristics within one specific model, including differences in land-sea distribution, topography, ice sheets, and lakes. While the former three have already been available from common fx variables, we made sure that the CMIP7 data request will also allow documentation of different lake distributions for research across and beyond CMIP.

Thus, the complexity of this question lies out of scope for our short paper.

Minor comments:

I.71: One could also cite IPCC AR6 WG1 cross-chapter box 10.1 here

Done.

1.75: It is unclear to me which data set the v2.2 refers to

Clarified.

I.76: Something seems wrong with this sentence

Fixed.

I.120: What about ice-sheet--ocean interactions, was ISMIP involved in these discussions? This seems like a topic that I found surprising to not be covered as a major topic/opportunity in this paper

This topic was discussed within our ocean team (particularly around ice shelf-ocean interaction), but as with the tipping point community, we did not receive an opportunity request from the community. The number of climate models where ice sheets are simulated online is still few, and where these data should be stored (with ocean and sea ice, in with hydrology/snow modeling, on their own) remains undecided at this point.

The Land and Land Ice Data Request team (Li, Y., Tang, G., O'Rourke, E., Minallah, S., e Braga, M. M., Nowicki, S., Smith, R. S., Lawrence, D. M., Hurtt, G. C., Peano, D., Meyer, G., Hassler, B., Mao, J., Xue, Y., and Juckes, M.: CMIP7 Data Request: Land and Land Ice Priorities and Opportunities, EGUsphere, https://doi.org/10.5194/egusphere-2025-3207, 2025.) have the primary responsibility for the ice sheet data request.

Table 1 (and other tables): Please check column width, in this table the first column is too narrow and so all IDs are spread over multiple lines

Until the formatting is done by EGU, we cannot get these precisely, but we will check carefully at the proof stage that all tables are correct and legible.

I.297: Please spell out AFT here, the abbreviation is rarely used in this paper and there are many pages after its definition in I.90. The mentioning of the 127k simulations is surprising, I don't understand what these refer to - maybe provide. a bit more background information for the non-paleo readership of this paper?

Fixed.

I.492: The style of this paragraph is different to the ones before, in particular owing to the usage of "we request". Might be good to streamline the style of all opportunities one way or the other.

Fixed.

I.501: Again, please spell out AFT I suggest.

Fixed.

Appendix B: sisnmassn and sisnmasss could use same wording for their titles

Fixed.

---

## Author Comment (AC2)

Key: Blue Text=Reviewer. Black Text=Author Response. Purple Text=Alterations to the manuscript.

**Reviewer #2: Brandon Reichl, GFDL**

This paper proposes the framework for ocean and sea-ice focused data output standards for CMIP7 models. The text provides a summary of this project's origination, structure, and recommendations. The objective of this manuscript is to document this committee's task to create a high-level overview for the recommended marine based CMIP7 data requests and summarize the process. This objective is achieved. Since the paper lays out an important set of recommendations for the CMIP7 community, it is appropriate for publishing. My comments can be viewed as suggestions that the authors may consider for revision rather than concerns that need to be addressed to warrant publication.

We thank Dr. Reichl for his time and effort in helping with this process.

**General Comments**

The authors clearly devoted considerable time and effort into generating this data request, and it is a monumental feat to achieve this summary. This paper could easily have been 2-3x its current length depending on the granularity of presentation of the opportunities and the discussion of output variables associated with each. Distilling it down to something more manageable is a positive outcome, but this does sacrifice the ability of this paper to be self-contained. As such, there is much jargon and detail that assumes a reader is already intimately familiar with CMIP and the CMIP7 data requests. The reader is also assumed to know where to find the details that were omitted. That is probably fine for the intended audience and for the purpose of this work to fit within a collection of related manuscripts. It is a very different paper from the OMIP protocol and diagnostics paper (Griffies et al., 2016), which seems fine given a more high-level purpose.

We appreciate the thoughtfulness of this remark and appreciate your suggestions for improvements where possible.

It took me a considerable amount of time to figure out how to find the data variables for this request.

I found the Airtable website (https://airtable.com/appOcSa4gXyzHThmm/shrkayKObes58Zu45/tbljoSaMlK7m0DunX/viw0ev RBr0vqp658c) after some exploring through the provided Zenodo link, github, and the binder software implementation, and eventually realized this was all organized on a website (https://wcrp-cmip.org/cmip7-data-request-v1-2-2/). Is the website address intentionally omitted from this text? The Airtable seemed to me the most direct way to scan through the details of the data request without installing special software and spending some time learning to navigate the data structures. Maybe it was omitted because it is not a permanent resource? Could some of the information be exported to PDF supplementary materials?

We intend to offer such permanent links alongside the paper. We will check with the CMIP-IPO and find out the planned implementation, and make it clear in the final version.

I do not notice any obvious omissions of ocean and sea-ice model output that would prevent CMIP7 models from serving their primary purposes. There are a number of non-baseline variables included whose necessity could be debated at length, and probably already was debated within this large author group. The discussion about why these variables are needed is laid out well scientifically. However, I do find the data justification of this request lacking in detail, focusing on these science questions that can always be better addressed with more data, but not reporting much detail on the consideration of specific data demands. Of course, judging the appropriateness of certain data volume requests will vary significantly depending on one's specific interests and the resources available to a specific modeling institution. If possible, providing some quantitative estimates related to the added data cost of various opportunities, variable groups, etc. would be helpful. This would especially help convey that this real (and significant) implication of the cost to institutions of serving the data request was weighed against the science it will support.

This is an important point. There are estimates of the data volume that can be extracted from the Airtable. We will find these estimates and include them in the paper. There are some companion papers in progress, such as the OMIP protocol paper, which will add more detail relevant to that narrower audience.

Prioritization of outputs (including variables or frequency) could also have been discussed more, if a goal is to ensure more institutions provide certain specific model output. There are some "high", "medium", and "low" priority cases, but what gives a variable its priority is not discussed in detail. Is it expected that modeling centers under numerous pressures from deadlines will be able to save and publish the "low" priority output? My takeaway from Section 4.1 is that a proper prioritization process was very difficult and likely demanded more resources than were available.

The prioritization was indeed an overwhelming task. The basic approach was to begin with the prioritization from the CMIP6 data request (Griffies et al., 2016; Notz et al., 2016) and then modify based on assuming a lower median level to begin, then raising and lowering the priorities based on our discussions and the download and upload statistics from CMIP6 archives. Prioritization is very subjective, and one may argue that many variables featured in a low-priority category may actually be high priority for a specific scientific purpose. We provide one perspective on variable priority, acknowledging that this is not the only point of view. Modelling centers will then have to decide what they can produce, depending on feasibility.

A discussion on spatial coarsening of model output could be useful somewhere. Some variables (e.g., heat content, certain scalars, MOC) can likely be coarsened with the savings in data serving demands outweighing the loss in information (e.g., from a ¼ degree grid to a 1 degree grid). But other data should presumably not be coarsened beyond the model native resolution if possible (e.g., extremes; winds if the purpose is for driving a wave model).

This is a good point, and in the CMIP6 archives, model variables were categorized into those that should be on the native grid (i.e., those primarily intended for budgets to close) and those that can be safely interpolated onto a 1 degree grid. However, there was no easy way within the

data request protocol to decimate data in the horizontal. Decimation in time and in the vertical were more easily done and were utilized by our team.

I wonder if "Waves" belong as their own topic alongside ocean and sea-ice? E.g., the title could be "ocean, sea-ice, and wave priorities and opportunities", given the attention to waves in this data request. This emphasis on waves would then better justify why two of the more data intensive opportunities are associated with waves parameters.

Given that waves are inherently part of the ocean realm, we believe the current title is sufficient to describe the extent of this data request. It is true that 2 out of the 7 opportunities relate to waves. This was particularly interesting to our team and not of our own design. A few of our authors were involved in wave modeling efforts for CMIP6 (including Fox-Kemper), but those opportunities were contributed from outside of our team. The fact that there are two opportunities reflects the two ways that waves are simulated in CMIP, offline via the COWCLIP team and its descendants and online via the efforts at FIO and NCAR during CMIP6 and additional teams joining in CMIP7. Because these two applications differ substantially in which data is requested (forcing vs. forcing and response, essentially), and the fact that only some modeling centers have online wave capabilities, while all modeling centers can choose to provide forcing data for offline wave calculations, we found it easier to organize them separately.

**Specific Comments**

Introduction

L77: Check this sentence, maybe replace "has" with "and".

Fixed.

L82: Multiple time [and spatial] scales. (is frequency needed here?)

Fixed.

L86: The terms in italics could be defined somewhere (perhaps in a table). E.g., after reading this paper a few times I'm still not entirely clear on what is meant by "opportunities".

These are terms of art that extend across the entire data request. However, we have confirmed that definitions are provided on first use.

L107: "The accompanying tables..." This reference seems to only include a small spreadsheet of the REF variables. Is it meant to be to the full opportunities/groups/variables discussed in this text?

As mentioned above, we will add links that point to a permanent archives of the full data lists.

**Approach and Methodology**

This essentially reads like a technical report, there is not much to comment on regarding the methodology.

Unavoidably so, unfortunately. We experimented with different language styles but none were major improvements.

Table 1: I wonder if there is a way to add more granularity to the breakdown of the variables. E.g., ID 47 has 240 variables, but if they are 2d monthly variables it would be a completely different request than if they are 240 3d daily mean variables. Grouping by time period and/or 2d vs 3d could help clarify the data demand. If data volumes were part of the decision making, it could also help to explicitly include some quantitative data estimates (e.g., many data volume estimates are given by Juckes et al., 2024). I eventually figured out that this information is contained in the AirTable (if that is reliable?), but without significant experience using AirTable that wasn't obvious to me and so could be summarized here.

The airtables are the correct repository, and sorting as you suggest is possible there. We will point to the permanent archives of these tables from the paper and note the capability for grouping and sorting there. We have produced a csv formatted version of the archive, which should be robust enough to avoid issues from future software changes in python, etc.

Table 1: "Experiment Groups" might be useful to define.

Fixed.

3.1, Ocean Changes, Drivers and Impacts

ENSO is briefly mentioned, but the science questions referenced here largely neglect tropical topics. Maybe that reflects the state of the scientific interests now, but perhaps some ENSO implications could be mentioned?

Added.

No glaring variable omissions. The total estimate of data volume on AirTable is about 23 TB, which seems manageable. The ocean\_mesoscale addition is a fairly substantial fraction of this, I think that their cost could be acknowledged.

Noted the high-res implications and estimated the total.

L183: What is the practical benefit of this clustering? What is the benefit of a variable group?

Now defined more carefully on first use.

Section 3.2 (Sea Ice Changes, Drivers, and Impacts)

No comments, the data estimate from AirTable was 16.1 TB

Thank you.

Section 3.3 (Paleoclimate)

The data estimate from AirTable was 22.4 TB

Thank you. There is some overlap with the preceding opportunities' lists.

L297: Some elaboration could help here, the assumption now is that a reader is familiar with paleoclimate experiments and knows what the abrupt-127k simulation means (I had to look it up)

A more extensive definition is now provided. We link to relevant publications that define abrupt-127k and that illustrate the function of this simulation in the context of the wider CMIP7 framework.

Section 3.4 (Polar Amplification):

The data estimate from AirTable was 35.8 TB

Thank you. There is some overlap with the preceding opportunities' lists.

Section 3.5 (Extremes):

The data estimate from AirTable was 54 TB, this opportunity will likely be an important one from CMIP7 and will be well utilized by groups performing model analysis.

Thank you. We agree that this opportunity will provide important new insights.

L386: I'm surprised to see BGC data also included in this request, it seems somewhat out of the stated scope.

This is only a limited set, selected here only for the purpose of multi-component extremes. These variables are treated in a larger BGC context by the biogeochemistry data request team.

Section 3.6 (Wind waves)

The total size of this data request on AirTable is 64 TB, or almost 3x that of the Ocean Changes opportunity. That significant potential overhead is probably worth discussing, or at least explaining how much of the data is an additional cost vs atmospheric variables that would already have been stored as part of other opportunities.

Thank you. We re-evaluated the experiments list that this opportunity points to and considered options to reduce the data footprint.

L405: Those that do are often at rather coarse grid spacing (this is sort of mentioned in other places).

Noted.

L407: "high-resolution data" -> This is subjective, it could be elaborated what time frequency and spatial resolution are preferred.

Specified.

L409: I'm unclear what is meant by "independent of ESM outputs". My understanding was that the ESM outputs are to be used to drive the wave model?

Clarified.

L411: Should it be obvious why this offers computationally efficiency and spatial detail? I'm not sure that it is.

Clarified.

Section 3.7 (wave coupling)

This is the most expensive opportunity in this topic area according to AirTable at 97 TB, or roughly 4x the size estimate of Ocean Changes. If the AirTable estimates are reliable and these are not just associated with otherwise collected variables I think some discussion is warranted. This opportunity being much more data intensive than the extreme impacts opportunity does make it feel like a substantial new request.

Thank you. We re-evaluated the list of experiments and variables for this opportunity. The new set of wave opportunity variables has been reduced to two groups (3-hourly and monthly). These changes should reduce the overall storage estimate compared with the previous version. However, it is important to note that not all model experiments are selected for this opportunity, we anticipate only a limited number of modeling centers will have active wave components. We also will propose (optional) time slices to provide these variables, which will prioritize a few time windows most meaningful for comparison of wave coupling impacts.

Appendix B: I'm unsure what the need is for choosing this subset of variables to define more formally (aside from being new to CMIP7?). It is a rather long/unwieldy list, but is incomplete and thus requires intimate knowledge of CMIP6 variables. A lot of non ocean and cryosphere variables are also included, is that intentional? I generally like the inclusion of some key variable information that can be accessed without surveying the AirTable (could be this appendix, or supplementary materials).

Yes, this list is provided to add some clarity (to the previously initiated) about the new variables. Although this list is long, we felt that it was helpful as it differs from preceding data requests. The non-ocean variables are relevant to the sea ice and paleoclimate opportunity, which is in scope (note that this is the ocean AND sea ice data request).

Typos/Word choice:

L163: As already noted [in the introduction and references within]

Fixed.

208: resolution -> grid-spacing

Fixed.

208: higher -> finer

**Fixed.**

Citation: https://doi.org/10.5194/egusphere-2025-3083-RC2